# Anomaly-Preference Image Generation

**Fuyun Wang** [1]   **Yuanzhi Wang** [1]   **Xu Guo** [2]   **Sujia Huang** [1]   **Tong Zhang** [1]   **Dan Wang** [3]   **Xin Liu** [4]   **Hui Yan** [1]   **Zhen Cui** [2]

## Abstract

Synthesizing realistic and diverse anomalous samples from limited data is vital for robust model generalization. However, existing methods struggle to reconcile fidelity and diversity, often hampered by distribution misalignment and overfitting, respectively. To mitigate this, we introduce Anomaly Preference Optimization (APO), a novel paradigm that reformulates anomaly generation as a preference learning problem. Central to our approach is an implicit preference alignment mechanism that leverages real anomalies as positive references, deriving optimization signals directly from denoising trajectory deviations without requiring costly human annotation. Furthermore, we propose a Time-Aware Capacity Allocation module that dynamically distributes model capacity along the diffusion timeline—prioritizing structural diversity during high-noise phases while enhancing fine-grained fidelity in low-noise stages. During inference, a hierarchical sampling strategy modulates the coherence-alignment trade-off, enabling precise control over generation. Extensive experiments demonstrate that significantly outperforms existing baselines, achieving state-of-the-art performance in both realism and diversity.

## 1. Introduction

Industrial anomaly detection is pivotal to applications ranging from automated visual inspection to medical imaging. However, the scarcity of anomalous samples coupled with

[1]Nanjing University of Science and Technology, Nanjing, China [2]Beijing Normal University, Beijing, China [3]China Academy of Space Technology, Beijing, China [4]Nanjing SeetaCloud Technology, Nanjing, China. Correspondence to: Tong Zhang <tong.zhang@njust.edu.cn>, Zhen Cui <zhen.cui@bnu.edu.cn>.

*Proceedings of the $43^{rd}$ International Conference on Machine Learning*, Seoul, South Korea. PMLR 306, 2026. Copyright 2026 by the author(s).

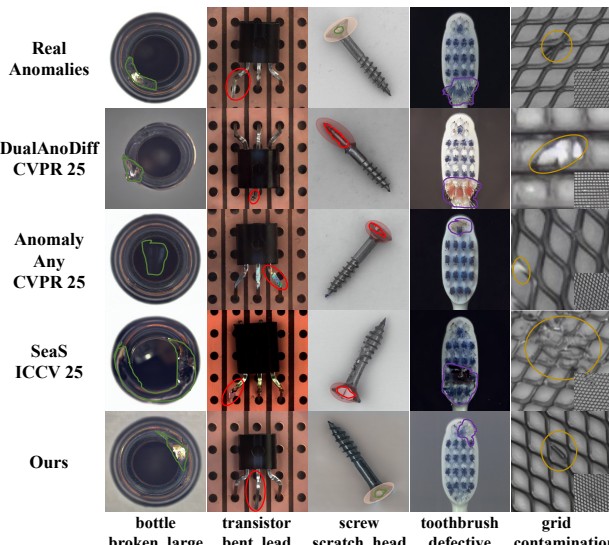

*Figure 1.* Compared with state-of-the-art methods including AnomalyDiffusion (Hu et al., 2024), DualAnoDiff (Jin et al., 2025), AnomalyAny (Sun et al., 2025) and SeaS (Dai et al., 2024), our approach have achieved superior performance.

the prohibitive cost of expert annotation severely constrains the model generalization to unseen defects. Recent methods (Sun et al., 2025; Dai et al., 2024) aim to synthesize realistic and diverse anomalies from sparse examples. This strategy effectively bolsters generalization in data-limited settings, offering substantial performance gains in open-set and low-resource detection scenarios (Wang et al., 2025).

Early anomaly generation relied on hand-crafted heuristics such as CutPaste (Li et al., 2021) to simulate defects. However, these approaches often yield semantically incoherent artifacts and fail to model the structural diversity of real anomalies. Leveraging the robust generative priors of diffusion models, recent research has achieved more plausible anomaly synthesis (Hu et al., 2024; Dai et al., 2024; Jin et al., 2025; Sun et al., 2025), primarily classifying into fine-tuning-based and training-free paradigms. Fine-tuning methods (Jin et al., 2025; Dai et al., 2024) adapt pre-trained models on limited data using textual inversion or conditional guidance to capture domain-specific patterns. In contrast, training-free methods (Sun et al., 2025) preserve the original parameters and synthesize anomalies solely during inference

by manipulating attention or feature maps.

Despite recent progress, balancing realism and diversity remains a persistent challenge. Fine-tuning-based approaches are frequently compromised by the decoupled learning of appearance and location, which induces semantic incoherence. Simultaneously, the implicit interactions within their dual-stream architectures often trigger feature conflicts, degrading structural integrity and leading to training instability due to gradient interference. Conversely, training-free methods shift the computational burden to the inference stage, creating a latency bottleneck that restricts their utility in resource-constrained environments. Critically, neither paradigm ensures explicit distributional alignment with target anomalies. While Direct Preference Optimization (Rafailov et al., 2023) offers a theoretical foundation for such alignment, its reliance on extensive human annotation renders it impractical for data-scarce scenarios. This motivates our core question: *How can we formulate a stable optimization objective that achieves direct distributional alignment via implicit preference learning, deriving supervision signals exclusively from limited anomaly examples?*

In response, we propose Anomaly Preference Optimization (APO). This unified framework reformulates few-shot anomaly generation as a constrained policy optimization task that requires no human annotation. Central to our approach is an implicit preference alignment mechanism. By leveraging the latent denoising trajectory deviations of real anomalies as intrinsic supervision signals, we enforce strict distributional alignment with target defects while preserving the generative priors of the base model. Furthermore, to reconcile the trade-off between diversity and fidelity, we introduce a Time-Aware Capacity Allocation (TACA) strategy. This module dynamically modulates the adaptation capacity of the model along the diffusion timeline by limiting updates during high-noise phases to maintain structural diversity while expanding capacity in low-noise stages to capture fine-grained patterns. Finally, we implement a Hierarchical Sampling scheme during inference. This strategy decouples semantic context from defect injection to offer precise control over the generation process. Extensive experiments demonstrate that APO not only synthesizes high-fidelity anomalies but also yields significant gains in downstream detection performance.

In summary, our contributions are three-fold:

- We propose APO, a novel framework for few-shot anomaly generation that leverages implicit preference alignment to eliminate explicit reward modeling or reinforcement learning.

- We design a time-aware capacity allocation and a hierarchical sampling strategy, which collectively balance structural diversity and fine-grained fidelity across the

diffusion trajectory.

- Extensive experiments show state-of-the-art performance in anomaly synthesis quality and downstream detection accuracy across multiple benchmarks.

## 2. Related Work

### 2.1. Anomaly Image Generation

Industrial anomaly detection suffers from inherent data scarcity due to the rarity of defects, motivating the development of anomaly generation techniques. Early model-free approaches like Cut-Paste (Li et al., 2021) synthesize anomalies through patch transplantation but produce unrealistic artifacts due to limited contextual modeling. GAN-based methods such as DFMGAN (Duan et al., 2023) adapt pretrained models with few anomalies yet struggle with semantic coherence.

Recent diffusion-based approaches can be categorized into fine-tuning and training-free paradigms. Fine-tuning methods like AnomalyDiffusion (Hu et al., 2024) often decouple appearance and location learning, resulting in semantically inconsistent generations. Dual-stream architectures such as DualAnoDiff (Jin et al., 2025) introduce feature conflicts that harm structural integrity, while unified models like SeaS (Dai et al., 2024) face training instability from complex multi-task optimization. Training-free alternatives like AnomalyAny (Sun et al., 2025) avoid fine-tuning but incur substantial inference overhead. These limitations collectively stem from the fundamental distributional misalignment between generated and real anomaly distributions, severely compromising synthesis realism.

### 2.2. Reinforcement Learning from Human Feedback

Reinforcement Learning from Human Feedback (RLHF) has become a standard approach for aligning models with human preferences, particularly in language modeling (Christiano et al., 2017; Bai et al., 2022a). The typical RLHF pipeline first trains a reward model on human-annotated preference pairs, then fine-tunes the base model using reinforcement learning objectives like PPO (Lee et al., 2023). However, this two-stage process faces computational overhead and training instability. Direct Preference Optimization (DPO) (Rafailov et al., 2023) addresses these issues by directly optimizing policies through a closed-form preference loss, eliminating explicit reward modeling. This stable framework has inspired extensions including SLiC-HF (Zhao et al., 2023) and ORPO (Hong et al., 2024) for improved preference alignment.

In computer vision, preference alignment has been increasingly applied to diffusion models for visual quality enhancement (Fan et al., 2023). Existing approaches either extend

PPO-based RLHF to image generation or adapt DPO using human-ranked images (Wallace et al., 2024). However, these methods remain dependent on costly human annotations and primarily target general quality improvement. Our work extends DPO to few-shot anomaly generation by replacing human-annotated preferences with implicit alignment signals from real anomaly samples, eliminating annotation requirements while enabling domain-specific distribution alignment.

## 3. Preminilaries

**Stable Diffusion.** We adopt the latent-space diffusion formulation of Stable Diffusion (Rombach et al., 2022). Given an image $\mathbf{x}$, an encoder $\mathcal{E}$ maps it to a latent $\mathbf{z}_0 = \mathcal{E}(\mathbf{x})$; with a text condition embedding $\mathbf{c}$, the forward noising process is $\mathbf{z}_t = \alpha_t \mathbf{z}_0 + \sigma_t \boldsymbol{\epsilon}$, where $t \sim \mathcal{U}(0, T)$, $\boldsymbol{\epsilon} \sim \mathcal{N}(\mathbf{0}, \mathbf{I})$, and $\alpha_t, \sigma_t$ is a noise schedule ($\alpha_t^2 + \sigma_t^2 = 1$). $\lambda_t = \log(\alpha_t^2/\sigma_t^2)$ denotes the log signal-to-noise ratio (log-SNR) at time $t$ and $\lambda_t'$ denotes the time-derivative of $\lambda_t$. A denoising network $\epsilon_\theta$ is trained to predict the injected noise in latent space using:

$$\mathcal{L}_{\text{SD}} = \mathbb{E}_{\mathbf{z}_0, \mathbf{c}, t, \boldsymbol{\epsilon}} \left[ \|\boldsymbol{\epsilon} - \epsilon_\theta(\mathbf{z}_t, \mathbf{c}, t)\|_2^2 \right]. \tag{1}$$

**Direct Preference Optimization.** In preference-based learning we assume access to condition-labeled comparison pairs $(\mathbf{z}_0^w, \mathbf{z}_0^l) \sim \mathcal{D}_z$ for a conditioning $\mathbf{c}$, with $\mathbf{z}_0^w \succ \mathbf{z}_0^l | \mathbf{c}$ indicating that $\mathbf{z}_0^w$ is preferred over $\mathbf{z}_0^l$. The Bradley-Terry model (Bradley & Terry, 1952) formulates the probability of preference as:

$$p_{\text{BT}}(\mathbf{z}_0^w \succ \mathbf{z}_0^l | \mathbf{c}) = \text{sigmoid} \left( \mathcal{R}(\mathbf{z}_0^w, \mathbf{c}) - \mathcal{R}(\mathbf{z}_0^l, \mathbf{c}) \right), \tag{2}$$

where $\text{sigmoid}(\cdot)$ is the sigmoid function and $\mathcal{R}(\cdot)$ represents an implicit reward function measuring sample quality. The reinforcement learning from human feedback (RLHF) (Bai et al., 2022b) framework aims to optimize a policy $p_\theta(\mathbf{z}_0 | \mathbf{c})$ to maximize expected reward while regularizing against a reference model $p_{\text{ref}}(\mathbf{z}_0 | \mathbf{c})$:

$$\max_{p_\theta} \mathbb{E}_{\mathbf{c} \sim \mathcal{D}_c, \mathbf{z}_0 \sim p_\theta(\mathbf{z}_0 | \mathbf{c})} [\mathcal{R}(\mathbf{z}_0, \mathbf{c})] - \beta \mathbb{D}_{\text{KL}} \left[ p_\theta(\mathbf{z}_0 | \mathbf{c}) \| p_{\text{ref}}(\mathbf{z}_0 | \mathbf{c}) \right], \tag{3}$$

where $\beta$ controls the regularization strength.

Direct Preference Optimization (DPO) (Rafailov et al., 2023) provides an elegant solution to solve Eqn. (3) by deriving the optimal policy form as follows:

$$p_\theta^*(\mathbf{z}_0 | \mathbf{c}) = \frac{1}{Z(\mathbf{c})} p_{\text{ref}}(\mathbf{z}_0 | \mathbf{c}) \exp \left( \frac{\mathcal{R}(\mathbf{z}_0, \mathbf{c})}{\beta} \right), \tag{4}$$

where $Z(\mathbf{c}) = \sum_{\mathbf{z}_0} p_{\text{ref}}(\mathbf{z}_0 | \mathbf{c}) \exp(\mathcal{R}(\mathbf{z}_0, \mathbf{c})/\beta)$ is the partition function. This leads to the key reparameterization:

$$\mathcal{R}(\mathbf{z}_0, \mathbf{c}) = \beta \log \frac{p_\theta^*(\mathbf{z}_0 | \mathbf{c})}{p_{\text{ref}}(\mathbf{z}_0 | \mathbf{c})} + \beta \log Z(\mathbf{c}). \tag{5}$$

The DPO objective which optimizes the policy is defined:

$$\mathcal{L}_{\text{DPO}} = -\mathbb{E}_{\mathbf{c}, \mathbf{z}_0^w, \mathbf{z}_0^l}$$
$$\left[ \log \text{sigmoid} \left( \beta \log \frac{p_\theta(\mathbf{z}_0^w | \mathbf{c})}{p_{\text{ref}}(\mathbf{z}_0^w | \mathbf{c})} - \beta \log \frac{p_\theta(\mathbf{z}_0^l | \mathbf{c})}{p_{\text{ref}}(\mathbf{z}_0^l | \mathbf{c})} \right) \right]. \tag{6}$$

While DPO provides a principled route to preference alignment, the absence of preference pairs in few-shot anomaly synthesis leads us to adopt a reparameterized objective that bypasses explicit reward learning and RL, directly optimizing a KL-regularized conditional policy $p_\theta(\mathbf{z}_0 \mid \mathbf{c})$ using an implicitly defined consistency reward $\mathcal{R}(\mathbf{z}_0, \mathbf{c})$.

## 4. Method

### 4.1. Problem Definition

In the few-shot anomaly generation settings, we are provided with a limited reference set $\mathcal{D}_{\text{ref}} = \{(\mathbf{z}_{i,0}, \mathbf{c}_i)\}_{i=1}^K$, where $\mathbf{z}_{i,0} = \mathcal{E}(\mathbf{x}_i)$ is the latent representation of an anomaly sample $\mathbf{x}_i$ with textual condition $\mathbf{c}_i$, subscript $i$ denotes the sample index and 0 denotes the diffusion timestep. Let $p_{\text{ref}}(\mathbf{z}_{i,0} | \mathbf{c}_i)$ denote reference distribution induced by a pretrained latent diffusion model. Our goal is to learn an anomaly-aligned generation policy $p_\theta(\mathbf{z}_{i,0} | \mathbf{c}_i)$ that synthesizes novel, high-fidelity anomaly samples consistent with the underlying structure and anomaly patterns in $\mathcal{D}_{\text{ref}}$, while retaining the general generative capability and diversity.

*Our abstract idea* formulates few-shot anomaly generation as a constrained policy optimization problem, where we aim to learn a generative policy $p_\theta(\mathbf{z}_0 | \mathbf{c})$ that maximizes alignment with target anomaly patterns while maintaining proximity to the reference distribution. Formally, we seek to optimize:

$$\max_\theta \quad \mathbb{E}_{\mathbf{c} \sim \mathcal{D}_{\text{ref}}, \mathbf{z}_0 \sim p_\theta(\mathbf{z}_0 | \mathbf{c})} [\mathcal{R}(\mathbf{z}_0, \mathbf{c})], \tag{7}$$

$$\text{s.t.} \quad \mathbb{E}_{\mathbf{c} \sim \mathcal{D}_{\text{ref}}} [D_{\text{KL}}(p_\theta(\mathbf{z}_0 | \mathbf{c}) \| p_{\text{ref}}(\mathbf{z}_0 | \mathbf{c}))] \leq \epsilon, \tag{8}$$

where $\mathcal{R}(\mathbf{z}_0, \mathbf{c})$ is an implicit reward function quantifying anomaly alignment, and $\epsilon > 0$ defines the maximum allowable distributional shift to preserve generation stability. For tractable optimization, we employ the Lagrangian dual formulation, which yields the practical objective:

$$\max_\theta \mathbb{E}_{\mathbf{c} \sim \mathcal{D}_{\text{ref}}, \mathbf{z}_0 \sim p_\theta(\mathbf{z}_0 | \mathbf{c})} [\mathcal{R}(\mathbf{z}_0, \mathbf{c})] - \beta \cdot \mathbb{E}_{\mathbf{c} \sim \mathcal{D}_{\text{ref}}} [D_{\text{KL}}(p_\theta(\mathbf{z}_0 | \mathbf{c}) \| p_{\text{ref}}(\mathbf{z}_0 | \mathbf{c}))], \tag{9}$$

with $\beta > 0$ serving as the regularization coefficient that explicitly controls the trade-off between anomaly alignment and distributional conservation. This formulation maintains theoretical rigor while enabling efficient optimization, as detailed in Algorithm 1.

---

**Algorithm 1** APO: Anomaly Preference Optimization

---

**Require:** Base diffusion model $\epsilon_{\text{ref}}$, few-shot anomaly samples $\{(\mathbf{x}_i, \mathbf{c}_i)\}$, regularization coefficient $\beta$

**Ensure:** Optimized anomaly generation policy $\epsilon_\theta$ with TACA mechanism

1: Initialize policy network $\epsilon_\theta$ with time-aware LoRA adaptation
2: **while** not converged **do**
3:     **Step 1: Sample training instance**
4:     Sample real anomaly $(\mathbf{x}_i, \mathbf{c}_i) \sim \mathcal{D}_{\text{ref}}$, timestep $t \sim \mathcal{U}(0, T)$, noise $\epsilon \sim \mathcal{N}(0, I)$
5:     Compute noisy latent: $\mathbf{z}_t = \alpha_t \mathcal{E}(\mathbf{x}_i) + \sigma_t \epsilon$
6:     **Step 2: Estimate implicit reward via denoising deviation**
7:     Compute reference denoising: $\hat{\epsilon}_{\text{ref}} = \epsilon_{\text{ref}}(\mathbf{z}_t, \mathbf{c}_i, t)$
8:     Compute policy denoising: $\hat{\epsilon}_\theta = \epsilon_\theta(\mathbf{z}_t, \mathbf{c}_i, t)$
9:     Calculate alignment deviation: $\Delta = \|\hat{\epsilon}_\theta - \epsilon\|_2^2 - \|\hat{\epsilon}_{\text{ref}} - \epsilon\|_2^2$ {$\Delta < 0$ indicates superior policy alignment}
10:     **Step 3: Optimize constrained objective**
11:     Compute time-adaptive weight: $\beta_t = -\frac{1}{2}\beta\lambda_t'$
12:     Optimize implicit preference objective: $\mathcal{L}_{\text{APO}} = -\log\sigma(-\beta_t\Delta)$ {Maximizes $\mathcal{R}$ while enforcing KL constraint via $\beta$}
13: **end while**

---

## 4.2. Constrained Policy Optimization

While the direct preference optimization (DPO) framework provides a solid theoretical foundation, its direct application to few-shot anomaly synthesis faces a critical limitation: traditional DPO requires expensive human-annotated preference pairs where both positive and negative samples are generated by the model. In contrast, we propose Constrained Policy Optimization (CPO) which bypasses human annotation by leveraging real anomalies as positives and deriving implicit negative examples from denoising trajectory deviations. Our key insight stems from the optimal policy form derived from the constrained objective Eqn. (4):

$$p_\theta^*(\mathbf{z}_{i,0:T}|\mathbf{c}_i) \propto p_{\text{ref}}(\mathbf{z}_{i,0:T}|\mathbf{c}_i)\exp(\mathcal{R}(\mathbf{z}_{i,0:T}, \mathbf{c}_i)/\beta), \quad (10)$$

which reveals that the optimal policy is a reward-weighted version of the base model. Rearranging terms yields the pivotal reparameterization:

$$\mathcal{R}(\mathbf{z}_{i,0:T}, \mathbf{c}_i) = \beta\log\frac{p_\theta^*(\mathbf{z}_{i,0:T}|\mathbf{c}_i)}{p_{\text{ref}}(\mathbf{z}_{i,0:T}|\mathbf{c}_i)} + \beta\log Z(\mathbf{c}_i), \quad (11)$$

where $Z(\mathbf{c}_i)$ is the partition function. We introduce an implicit comparison mechanism that evaluates the relative denoising quality between the current policy and the reference model on shared real anomalies. This constructs a crucial preference signal: superior denoising directly indicates stronger alignment with the target anomaly distribution.

Formally, for a diffusion trajectory $\mathbf{z}_{i,0:T}$, we derive a tractable deviation function:

$$\delta(p_\theta, p_{\text{ref}}; \mathbf{x}_i, \mathbf{c}_i) = D_{\text{KL}}(q(\mathbf{z}_{i,0:T}|\mathbf{x}_i)\|p_{\text{ref}}(\mathbf{z}_{i,0:T}|\mathbf{c}_i)) - D_{\text{KL}}(q(\mathbf{z}_{i,0:T}|\mathbf{x}_i)\|p_\theta(\mathbf{z}_{i,0:T}|\mathbf{c}_i)). \quad (12)$$

where a positive $\delta > 0$ signifies stronger alignment with the target patterns, a property essential for preserving realism in few-shot anomaly generation. Via variational inference, we establish the following relation between the expected reward and the deviation function:

$$\mathbb{E}_{q(\mathbf{z}_{i,0:T}|\mathbf{x}_i)}[\mathcal{R}(\mathbf{z}_{i,0:T}, \mathbf{c}_i)] = \beta \cdot \delta(p_\theta, p_{\text{ref}}; \mathbf{x}_i, \mathbf{c}_i) + C, \quad (13)$$

where $C = \beta\log Z(\mathbf{c}_i)$ is a constant independent of $\mathbf{x}_i$. This result certifies that maximizing the deviation $\delta$ directly corresponds to maximizing the expected implicit reward, thereby enhancing generation realism. However, computing the exact deviation $\delta$ in Eqn. (12) requires evaluating likelihoods over the entire diffusion trajectory $\mathbf{z}_{i,0:T}$, which is computationally intractable.

To this end, we factorize the ELBO to obtain a tractable surrogate objective and, under the noise-prediction parameterization in Eqn. (1), derive a closed-form latent-space deviation function:

$$\delta(p_\theta, p_{\text{ref}}; \mathbf{x}_i, \mathbf{c}_i) = \frac{1}{2}\mathbb{E}_{t\sim\mathcal{U}(0,T),\epsilon\sim\mathcal{N}(0,\mathbf{I})}$$
$$\left[\lambda_t'\left(\|\epsilon_\theta(\mathbf{z}_{i,t};\mathbf{c}_i,t) - \epsilon\|_2^2 - \|\epsilon_{\text{ref}}(\mathbf{z}_{i,t};\mathbf{c}_i,t) - \epsilon\|_2^2\right)\right], \quad (14)$$

This formulation allows efficient Monte Carlo estimation during training by sampling random timesteps $t$ and noise vectors $\epsilon$. Substituting the tractable deviation expression Eqn. (14) into the reward reparameterization Eqn. (12), and applying the log-Sigmoid transformation to stabilize optimization, we derive our final anomaly-aligned objective:

$$\mathcal{L}_{\text{APO}} = \mathbb{E}_{t\sim\mathcal{U}(0,T),\,\epsilon\sim\mathcal{N}(\mathbf{0},\mathbf{I})}\Big[-\log\text{sigmoid}$$
$$\Big(-\beta_t\big(\|\epsilon_\theta(\mathbf{z}_{i,t};\mathbf{c}_i,t) - \epsilon\|_2^2$$
$$- \|\epsilon_{\text{ref}}(\mathbf{z}_{i,t};\mathbf{c}_i,t) - \epsilon\|_2^2\big)\Big)\Big], \quad (15)$$

where $\beta_t = -\frac{1}{2}\beta\lambda_t'$. $\mathcal{L}_{\text{APO}}$ provides a practical implementation that directly optimizes the implicit consistency reward while regularizing against the base model through the implicit KL divergence constraint, ensuring stable optimization without explicit negative examples.

## 4.3. Time-Aware Capacity Allocation

While our constrained policy optimization successfully enhances anomaly fidelity, it inadvertently suppresses generation diversity. The uniform fine-tuning inherent in this

process causes high-noise steps—responsible for structural layout—to overfit to background patterns, thereby limiting the model's capacity for structural variation. To mitigate this, we propose Time-Aware Capacity Allocation (TACA), which schedules the LoRA rank to be minimal at high-noise steps to preserve structural diversity, and maximal at low-noise steps to enhance fine-grained fidelity. Specifically, we implement time-aware capacity allocation through a temporal gating mechanism for low-rank adaptations. The weight update at each timestep is formulated as $\Delta W_t = B \cdot G_t \cdot A$, where the temporal gate $G_t$ modulates parameter influence through a dimension-selection mask:

$$G_t = \text{diag}(\underbrace{1, \ldots, 1}_{k(t)}, \underbrace{0, \ldots, 0}_{r-k(t)}), \qquad (16)$$

with the active dimension count $k(t)$ following an expansion schedule $k(t) = \lfloor k_{\min} + (k_{\max} - k_{\min}) \cdot (T-t)/T \rfloor$. This ensures minimal adaptation at high-noise steps $(t \to T)$ to preserve structural diversity, while progressively engaging more parameters at low-noise steps $(t \to 0)$ to enhance fine-grained anomaly synthesis.

### 4.4. Hierarchical Sampling Strategy

During inference, we introduce a hierarchical guidance sampling strategy to enable precise control over the trade-off between base model coherence and anomaly pattern consistency. This approach addresses the challenge of balancing the general generative capability of the reference model with the specific alignment to anomaly characteristics learned from few-shot examples.

The key insight is that the denoising process can be decomposed into three complementary components: the unconditional generation prior, text-conditioned semantic guidance, and anomaly-specific pattern injection. Formally, the denoising process is structured as:

$$\hat{\epsilon}(\mathbf{z}_{i,t}; \mathbf{c}_i, t) = \epsilon_{\text{ref}}(\mathbf{z}_{i,t}, t) + s_{\text{text}} \cdot \Delta_{\text{text}} + s_{\text{align}} \cdot \Delta_{\text{align}}, \quad (17)$$

where $\Delta_{\text{text}} = \epsilon_{\text{ref}}(\mathbf{z}_{i,t}; \mathbf{c}_i, t) - \epsilon_{\text{ref}}(\mathbf{z}_{i,t}, t)$ governs the text-conditioning strength to maintain semantic consistency, and $\Delta_{\text{align}} = \epsilon_{\theta}(\mathbf{z}_{i,t}; \mathbf{c}_i, t) - \epsilon_{\text{ref}}(\mathbf{z}_{i,t}; \mathbf{c}_i, t)$ regulates the anomaly pattern alignment strength to enhance few-shot adaptation. This formulation is mathematically equivalent to sampling from a guided distribution:

$$p_{\text{guided}}(\mathbf{z}_{i,t-1}|\mathbf{z}_{i,t}) \propto p_{\text{ref}}(\mathbf{z}_{i,t-1}|\mathbf{z}_{i,t}) \cdot$$
$$\left( \frac{p_{\text{ref}}(\mathbf{z}_{i,t-1}|\mathbf{z}_{i,t}, \mathbf{c}_i)}{p_{\text{ref}}(\mathbf{z}_{i,t-1}|\mathbf{z}_{i,t})} \right)^{s_{\text{text}}} \cdot \left( \frac{p_{\theta}(\mathbf{z}_{i,t-1}|\mathbf{z}_{i,t}, \mathbf{c}_i)}{p_{\text{ref}}(\mathbf{z}_{i,t-1}|\mathbf{z}_{i,t}, \mathbf{c}_i)} \right)^{s_{\text{align}}}.$$
$$(18)$$

*Deviation-Guided Anomaly Localization.* The alignment deviation $\Delta_{\text{align}} = \epsilon_{\theta}(\mathbf{z}_t; \mathbf{c}, t) - \epsilon_{\text{ref}}(\mathbf{z}_t; \mathbf{c}, t)$ in latent space

provides anomaly-sensitive signals, but requires careful mapping to pixel space for accurate localization. We address this through a multi-scale aggregation strategy:

$$\mathbf{M} = \frac{1}{T} \sum_{t=1}^{T} k(t) \cdot \text{Upsample}\left( \|\Delta_{\text{align}}(\mathbf{z}_t)\|_2 \right), \qquad (19)$$

where $\text{Upsample}(\cdot)$ denotes bilinear upsampling to match the spatial dimensions of the original image, and $k(t)$ follows the TACA expansion schedule. This direct spatial mapping preserves the relative anomaly strength while maintaining computational efficiency.

To enhance localization precision, we leverage the structural consistency between latent and pixel spaces:

$$\mathbf{P}_{\text{anomaly}} = \text{Smooth}\left( \frac{\mathbf{M} - \min(\mathbf{M})}{\max(\mathbf{M}) - \min(\mathbf{M})} \right), \qquad (20)$$

where $\text{Smooth}(\cdot)$ applies a fixed $3 \times 3$ Gaussian filter to reduce artifacts from the upsampling process while preserving anomaly boundaries. The resulting probability map directly reflects anomaly likelihood at pixel level without introducing domain-specific assumptions.

*The complete derivation of the relevant formulas mentioned above is included in the appendix.*

## 5. Experiment

### 5.1. Datasets

Our experiments are conducted on the MVTec benchmark (Bergmann et al., 2019), which contains 5,354 high-resolution images from 15 industrial categories (10 objects and 5 textures). Following the standard split, the dataset includes 3,629 normal images for training and 1,725 test images containing both normal and anomalous samples. Following AnomalyDiffusion (Hu et al., 2024), we use $1/3$ of anomalies per category as reference set, reserving the remaining $2/3$ for evaluation.

### 5.2. Evaluation Metrics

For generation assessment, we employ Inception Score (IS) to evaluate sample fidelity and Intra-cluster pairwise LPIPS distance (IC-LPIPS) to measure diversity; for anomaly inspection, we report AUROC, Average Precision (AP), and F1-max scores to quantify detection and localization accuracy across image and pixel levels.

### 5.3. Implementation Details

We initialize APO with the pre-trained weights from Stable Diffusion v1-4 (Rombach et al., 2022). We employ LoRA fine-tuning with TACA, setting the rank bounds to $k_{\min} = 4$ and $k_{\max} = 32$. The model is optimized using

*Table 1.* Quantitative comparison of IS and IC-LPIPS on MVTec. Our method achieves the best scores in both metrics. **Bold** and underlined values indicate the top and second-best results, respectively. AnomalyDiff: AnomalyDiffusion.

| Category | Crop&Paste IS ↑ | Crop&Paste IC-L ↑ | DFMGAN IS ↑ | DFMGAN IC-L ↑ | AnomalyDiff IS ↑ | AnomalyDiff IC-L ↑ | DualAnoDiff IS ↑ | DualAnoDiff IC-L ↑ | AnomalyAny IS ↑ | AnomalyAny IC-L ↑ | SeaS IS ↑ | SeaS IC-L ↑ | Ours IS ↑ | Ours IC-L ↑ |
|---|---|---|---|---|---|---|---|---|---|---|---|---|---|---|
| bottle | 1.43 | 0.04 | 1.62 | 0.12 | 1.58 | 0.19 | 2.17 | 0.43 | 1.73 | 0.17 | 1.78 | 0.21 | **2.19** | **0.45** |
| cable | 1.74 | 0.25 | 1.96 | 0.25 | 2.13 | 0.41 | 2.15 | 0.43 | 2.06 | 0.41 | 2.09 | 0.42 | **2.20** | **0.45** |
| capsule | 1.23 | 0.05 | 1.59 | 0.11 | 1.59 | 0.21 | 1.62 | 0.32 | 2.16 | 0.23 | 1.56 | 0.26 | **2.18** | **0.34** |
| carpet | 1.17 | 0.11 | 1.23 | 0.13 | 1.16 | 0.24 | 1.36 | 0.29 | 1.10 | 0.34 | 1.13 | 0.25 | **1.39** | **0.36** |
| grid | 2.00 | 0.12 | 1.97 | 0.13 | 2.04 | 0.44 | 2.13 | 0.42 | 2.31 | 0.38 | 2.43 | 0.44 | **2.44** | **0.46** |
| hazelnut | 1.74 | 0.21 | 1.93 | 0.24 | 2.13 | 0.31 | 1.94 | 0.35 | **2.55** | 0.32 | 1.87 | 0.31 | 2.07 | **0.37** |
| leather | 1.47 | 0.14 | 2.06 | 0.17 | 1.94 | 0.41 | 1.91 | 0.35 | 2.26 | 0.41 | 2.03 | 0.40 | **2.28** | **0.42** |
| metal_nut | 1.56 | 0.15 | 1.49 | 0.32 | 1.96 | 0.30 | 1.57 | 0.32 | 1.82 | 0.27 | 1.64 | 0.31 | **1.99** | **0.36** |
| pill | 1.49 | 0.11 | 1.63 | 0.16 | 1.61 | 0.26 | 1.82 | 0.38 | **2.91** | 0.30 | 1.62 | 0.33 | 2.55 | **0.40** |
| screw | 1.12 | 0.16 | 1.12 | 0.14 | 1.28 | 0.30 | 1.43 | 0.36 | 1.33 | 0.32 | 1.52 | 0.31 | **1.53** | **0.38** |
| tile | 1.83 | 0.20 | 2.39 | 0.22 | 2.54 | **0.55** | 2.40 | 0.50 | 2.66 | 0.53 | 2.60 | 0.50 | **2.68** | 0.54 |
| toothbrush | 1.30 | 0.08 | 1.82 | 0.18 | 1.68 | 0.21 | 2.40 | 0.48 | 1.64 | 0.22 | 1.96 | 0.25 | **2.45** | **0.49** |
| transistor | 1.39 | 0.15 | 1.64 | 0.25 | 1.57 | 0.34 | 1.71 | 0.33 | 1.66 | 0.28 | 1.51 | 0.34 | **1.77** | **0.36** |
| wood | 1.95 | 0.23 | 2.12 | 0.35 | 2.33 | 0.37 | 2.24 | 0.40 | 1.93 | 0.41 | **2.77** | **0.46** | 2.32 | 0.44 |
| zipper | 1.23 | 0.11 | 1.29 | 0.27 | 1.39 | 0.25 | 2.14 | 0.37 | 2.14 | 0.33 | 1.63 | 0.30 | **2.16** | **0.39** |
| average | 1.54 | 0.14 | 1.72 | 0.20 | 1.80 | 0.32 | 1.93 | 0.38 | 2.02 | 0.33 | 1.88 | 0.34 | **2.14** | **0.41** |

Adam (Kingma, 2014) with a learning rate of $5 \times 10^{-5}$ and batch size 1. For inference, we use the DDIM (Song et al., 2020) scheduler with 100 sampling steps and set the guidance scale $s_{\text{text}} = 6.5$ and $s_{\text{align}} = 3$. All experiments are conducted on NVIDIA GeForce RTX 4090 48G GPUs. In the following experiments, we generate 1,000 image-mask pairs per category to evaluate both generation quality and downstream inspection performance.

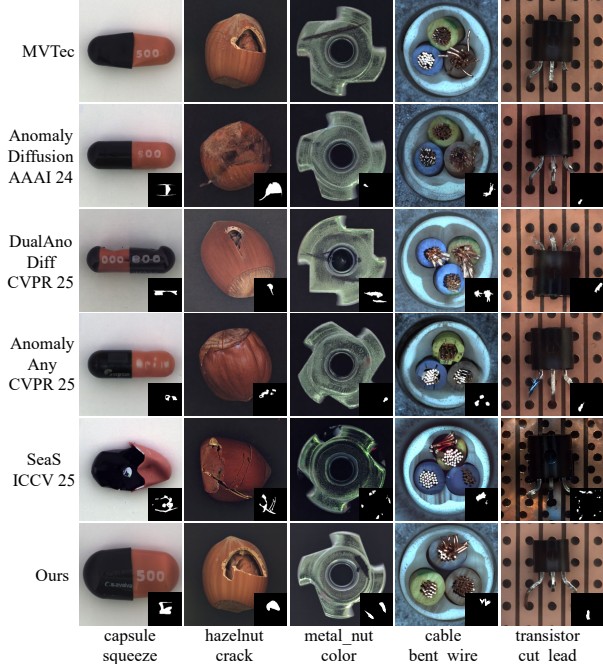

*Figure 2.* Comparative analysis on the MVTec dataset demonstrates our model's capability in generating high-quality anomaly images that faithfully reflect the provided masks.

## 5.4. Anomaly Generation Quality Comparison

**Baselines.** We evaluate our model against several established methods, namely Crop&Paste (Lin et al., 2021), DFMGAN (Duan et al., 2023), AnomalyDiff (Hu et al., 2024), DualAnoDiff (Jin et al., 2025), AnomalyAny (Sun et al., 2025), and SeaS (Dai et al., 2024).

**Anomaly Generation Quality.** Fig. 2 qualitatively demonstrates our method's superiority in generating faithful and semantically coherent anomalies. For structural defects like capsule–squeeze and transistor (cut_lead), where baselines often preserve original shapes or produce blur artifacts, our approach yields localized deformations that align with mask regions while preserving overall structure. For hazelnut–crack, baseline methods often over-distort the entire nut into implausible shapes, whereas our model preserves the global structure and only introduces a localized, mask-aligned crack. Similar structural artifacts appear in metal_nut (color) and cable (bent_wire), where competing approaches cause global color or geometry drifts, while our model yields realistic anomaly images.

As shown in Tab. 1, our our APO achieves state-of-the-art performance, improving average IS by 5.9% and IC-LPIPS by 7.9% over the strongest baseline. The most significant gains are observed in categories requiring precise structural control—such as zipper, cable, and bottle—where our method achieves superior realism while maintaining diversity. For texture-oriented categories like carpet and leather, we obtain balanced improvements in both metrics, demonstrating effective pattern preservation during anomaly synthesis. These results demonstrate our method's effectiveness in balancing realism and diversity.

**Anomaly Generation for Anomaly Detection and Localization.** Following established benchmarks, we compare

*Table 2.* Anomaly Detection Performance on MVTec AD. We compare the results of training a U-Net on synthetic data from DFMGAN, AnomalyDiffusion, DulAnoDiff, SeaS and our model for both pixel-level localization and image-level detection.

| Category | DFMGAN | | | | AnomalyDiffusion | | | | DualAnoDiff | | | | SeaS | | | | Ours | | | |
|---|---|---|---|---|---|---|---|---|---|---|---|---|---|---|---|---|---|---|---|---|
| | AUC-P | AP-P | $F_1$-P | AP-I | AUC-P | AP-P | $F_1$-P | AP-I | AUC-P | AP-P | $F_1$-P | AP-I | AUC-P | AP-P | $F_1$-P | AP-I | AUC-P | AP-P | $F_1$-P | AP-I |
| bottle | 98.9 | 90.2 | 83.9 | 99.8 | 99.4 | 94.1 | 87.3 | 99.9 | 99.5 | 93.4 | 85.7 | **100** | 99.7 | 95.9 | **88.9** | 99.9 | 99.6 | **96.5** | 88.2 | **100** |
| cable | 97.2 | 81.0 | 75.4 | 97.8 | **99.2** | **90.8** | **83.5** | **100** | 97.5 | 82.6 | 76.9 | 98.3 | 96.0 | 83.1 | 77.7 | 98.8 | 98.9 | 85.3 | 79.2 | 98.5 |
| capsule | 79.2 | 26.0 | 35.0 | 98.5 | 98.8 | 57.2 | 59.8 | **99.9** | 99.5 | 73.2 | 67.0 | 99.2 | 93.7 | 41.9 | 47.3 | 99.2 | **99.7** | **75.6** | **69.0** | 99.4 |
| carpet | 90.6 | 33.4 | 38.1 | 98.5 | 98.6 | 81.2 | 74.6 | 98.8 | 99.4 | 89.1 | 80.2 | 99.9 | 99.3 | 86.4 | 78.1 | 99.0 | **99.8** | **92.1** | **82.5** | **100** |
| grid | 75.2 | 14.3 | 20.5 | 90.4 | 98.3 | 52.9 | 54.6 | 99.5 | 98.5 | 57.2 | 54.9 | 99.7 | **99.7** | **76.3** | **70.0** | 99.9 | 99.0 | 60.1 | 57.2 | 99.8 |
| hazelnut | 99.7 | 95.2 | 89.5 | **100** | 99.8 | 96.5 | 90.6 | 99.9 | 99.8 | 97.7 | 92.8 | **100** | 99.5 | 92.3 | 85.6 | 99.8 | **99.9** | **98.5** | **94.0** | **100** |
| leather | 98.5 | 68.7 | 66.7 | **100** | 99.8 | 79.6 | 71.0 | **100** | **99.9** | **88.8** | 78.8 | **100** | 99.8 | 85.2 | 77.0 | **100** | **99.9** | **91.7** | **81.1** | **100** |
| metal_nut | 99.3 | 98.1 | 94.5 | 99.8 | **99.8** | 98.7 | 94.0 | **100** | 99.6 | 98.0 | 93.0 | 99.9 | **99.8** | 99.2 | **95.7** | **100** | **99.8** | **100** | **95.7** | **100** |
| pill | 81.2 | 67.8 | 72.6 | 91.7 | 99.8 | 97.0 | 90.8 | 99.6 | 99.6 | 95.8 | 89.2 | 99.0 | **99.9** | 97.1 | 90.7 | 99.6 | **99.9** | **97.5** | **92.0** | 99.8 |
| screw | 58.8 | 2.2 | 5.3 | 64.7 | 97.0 | 51.8 | 50.9 | 97.9 | 98.1 | 57.1 | 56.1 | 95.0 | 98.5 | 58.5 | 57.2 | **98.0** | **98.8** | **59.5** | **57.7** | 95.7 |
| tile | 99.5 | 97.1 | 91.6 | **100** | 99.2 | 93.9 | 86.2 | **100** | 99.7 | 97.1 | 91.0 | **100** | **99.8** | 97.9 | 92.5 | **100** | **99.8** | **99.3** | **93.7** | **100** |
| toothbrush | 96.4 | 75.9 | 72.6 | **100** | 99.2 | 76.5 | 73.4 | **100** | 98.2 | 68.3 | 68.6 | 99.7 | 98.4 | 70.0 | 68.1 | **100** | **99.5** | **77.8** | **74.6** | **100** |
| transistor | 96.2 | 81.2 | 77.0 | 92.5 | 99.3 | 92.6 | 85.7 | **100** | 98.0 | 86.7 | 79.6 | 99.7 | 98.0 | 87.3 | 81.9 | 99.5 | **99.6** | **93.8** | **87.0** | **100** |
| wood | 95.3 | 70.7 | 65.8 | 99.4 | 99.8 | 84.6 | 74.5 | 99.4 | 99.4 | 91.6 | 83.8 | 99.9 | 99.0 | 87.0 | 79.6 | 99.6 | **99.9** | **94.7** | **87.2** | **100** |
| zipper | 92.9 | 65.6 | 64.9 | 99.9 | 99.4 | 86.0 | 79.2 | **100** | 99.6 | 90.7 | 82.7 | **100** | 99.7 | 88.2 | 81.6 | **100** | 99.6 | **93.7** | **85.1** | **100** |
| Average | 90.0 | 62.7 | 62.1 | 94.8 | 99.1 | 81.4 | 76.3 | **99.7** | 99.1 | 84.5 | 78.8 | 98.9 | 98.7 | 83.1 | 78.1 | 99.6 | **99.6** | **87.7** | **81.6** | 99.5 |

*Table 3.* Anomaly Classification Accuracy (%) with ResNet-34 on Generated Data. AnoDiff: AnomalyDiffusion, DualAno:DualAnoDiff

| Category | DFMGAN | AnoDiff | DualAno | SeaS | Ours |
|---|---|---|---|---|---|
| bottle | 56.59 | **88.37** | 67.44 | 81.40 | 84.63 |
| cable | 45.31 | 76.56 | 57.81 | 48.44 | **83.37** |
| capsule | 37.23 | 44.00 | 50.67 | 33.33 | **57.87** |
| carpet | 47.31 | 58.06 | 62.90 | 38.71 | **63.35** |
| grid | 40.83 | 60.00 | **67.50** | 47.50 | 64.54 |
| hazelnut | **81.94** | 81.25 | 79.17 | 81.25 | 80.68 |
| leather | 49.73 | 65.08 | 84.13 | 61.90 | **86.52** |
| metal_nut | 64.58 | **82.81** | 73.44 | 68.75 | 78.48 |
| pill | 29.52 | 64.58 | 41.67 | 18.75 | **65.81** |
| screw | 37.45 | 29.63 | 39.51 | 56.79 | **58.95** |
| tile | 74.85 | 92.98 | **100** | 89.47 | **100** |
| transistor | 52.38 | 75.00 | 78.57 | 57.14 | **80.69** |
| wood | 49.21 | 78.57 | **90.48** | 76.19 | 88.32 |
| zipper | 27.64 | 86.59 | 24.39 | 34.15 | **87.74** |
| Average | 49.61 | 70.25 | 65.55 | 56.70 | **72.06** |

against DFMGAN, AnomalyDiffusion, and DualAnoDiff using results reported in their original papers, while AnomalyAny and SeaS are evaluated using official implementations or pre-trained models. As demonstrated in Table 2, the U-Net model trained on the anomalies generated by our method achieves state-of-the-art performance, attaining the highest AUC-P of 99.6% and an F1-max score of 81.6%. Besides, our approach also achieves competitive results across other metrics, including 87.7% AP-P and 99.5% AP-I at the image level. Notable performance gains are observed on challenging categories such as screw, capsule, and carpet, further validating the method's capability in generating structurally complex and subtle anomalies.

**Anomaly Generation for Anomaly Classification.** We further evaluate the semantic discriminability of generated anomalies through classification tasks using a ResNet-34 backbone. As summarized in Tab. 3, our method achieves the highest average classification accuracy of 72.06%, outperforming all compared generation approaches. The performance advantage is particularly pronounced in challenging categories like metal_nut and pill. Notably, in texture-rich categories including leather and tile, our method achieves near-perfect accuracy, indicating its effectiveness in maintaining both structural integrity and semantic consistency during anomaly synthesis.

## 5.5. Comparison with Anomaly Detection Models

We compare our method with state-of-the-art anomaly detection approaches, covering both unsupervised and supervised paradigms. The unsupervised baselines include DRAEM (Zavrtanik & Kristan, 2021), SSPCAB (Ristea et al., 2022), CFA (Lee et al., 2022), RD4AD (Deng & Li, 2022), and PatchCore (Roth et al., 2022), while the supervised counterparts include MuSc (Li et al., 2024), DevNet (Pang et al., 2021), DRA (Ding et al., 2022), and PRN (Zhang et al., 2023). Following standard evaluation protocols, we utilize official implementations for all baselines, except PRN where we report results from its original paper. As summarized in Tab. 4, our approach achieves superior performance with an average pixel-level AUROC of 99.6% and F1-max of 81.6%, establishing new state-of-

*Table 4.* Quantitative Comparison of Anomaly Localization Performance (AUROC/AP). We evaluate a standard U-Net trained on our generated anomalies against existing detection methods using official implementations or pre-trained models.

| Category | Unsupervised | | | | | | Supervised | | | |
|---|---|---|---|---|---|---|---|---|---|---|
| | DRAEM | SSPCAB | CFA | RD4AD | PatchCore | Musc | DevNet | DRA | PRN | Ours |
| bottle | 99.1/88.5 | 98.9/88.6 | 98.9/50.9 | 98.8/51.0 | 97.6/75.0 | 98.5/82.8 | 96.7/67.9 | 91.7/41.5 | 99.4/92.3 | **99.6/96.5** |
| cable | 94.8/61.4 | 93.1/52.1 | 98.4/79.8 | 98.8/77.0 | 96.8/65.9 | 96.2/58.8 | 97.9/67.6 | 86.1/34.8 | 98.8/78.9 | **98.9/85.3** |
| capsule | 97.6/47.9 | 90.4/48.7 | 98.9/71.1 | 99.0/60.5 | 98.6/46.6 | 98.9/52.7 | 91.1/46.6 | 88.5/11.0 | 98.5/62.2 | **99.7/75.6** |
| carpet | 96.3/62.5 | 92.3/49.1 | 99.1/47.7 | 99.4/46.0 | 98.7/65.0 | 99.4/75.3 | 94.6/19.6 | 98.2/54.0 | 99.0/82.0 | **99.8/92.1** |
| grid | 99.5/53.2 | **99.6**/58.2 | 98.6/**82.9** | 98.0/75.4 | 97.2/23.6 | 98.6/37.0 | 90.2/44.9 | 86.2/28.6 | 98.4/45.7 | 99.0/60.1 |
| hazelnut | 99.5/88.1 | 99.6/94.5 | 98.5/80.2 | 94.2/57.2 | 97.6/55.2 | 99.3/74.4 | 76.9/46.8 | 88.2/20.3 | 99.7/93.8 | **99.9/98.5** |
| leather | 98.8/68.5 | 97.2/60.3 | 96.2/60.9 | 96.6/53.5 | 98.9/43.4 | 99.7/62.6 | 94.3/66.2 | 80.7/5.1 | 99.7/69.7 | **99.9/91.7** |
| metal_nut | 98.7/91.6 | 99.3/95.1 | 98.6/74.6 | 97.3/53.8 | 97.5/86.8 | 87.5/49.6 | 93.5/57.4 | 80.3/20.6 | 99.7/98.0 | **99.8/100** |
| pill | 97.7/44.8 | 96.5/48.1 | 98.6/67.9 | 98.4/58.1 | 97.0/75.9 | 97.0/65.6 | 98.9/79.9 | 79.6/22.1 | 99.5/91.3 | **99.9/97.5** |
| screw | 99.7/**72.9** | 99.1/62.0 | 98.7/61.4 | 99.1/51.8 | 98.7/34.2 | **99.8**/31.7 | 66.5/21.1 | 51.0/5.1 | 97.5/44.9 | 98.8/59.5 |
| tile | 99.4/96.4 | 99.2/96.3 | 98.6/92.6 | 99.7/78.2 | 99.4/56.0 | 99.3/80.6 | 88.7/63.9 | 91.0/54.4 | 99.6/96.5 | **99.8/99.3** |
| toothbrush | 97.3/49.2 | 97.5/38.9 | 98.4/61.7 | 99.0/63.1 | 97.6/37.1 | 99.4/64.2 | 96.3/52.4 | 74.5/4.8 | **99.6/78.1** | 99.5/77.8 |
| transistor | 92.2/56.0 | 85.3/36.5 | 96.8/82.8 | **99.6**/50.3 | 91.8/66.7 | 92.5/61.2 | 55.2/4.4 | 79.3/11.2 | 98.4/85.6 | 99.6/93.8 |
| wood | 97.6/81.6 | 97.2/77.1 | 99.6/75.6 | 99.3/39.1 | 95.7/54.3 | 98.7/77.5 | 92.4/7.9 | 89.2/10.2 | 97.8/82.6 | **99.9/94.7** |
| zipper | 98.6/73.6 | 98.1/78.2 | 98.9/53.9 | 99.7/52.7 | 98.5/63.1 | 98.4/64.2 | 92.4/53.1 | 96.8/42.3 | 98.8/77.6 | **99.6/93.7** |
| Average | 97.7/69.0 | 96.2/65.5 | 98.3/66.3 | 98.3/57.8 | 97.1/56.6 | 97.5/62.6 | 86.4/49.3 | 84.8/25.7 | 99.0/78.6 | **99.6/87.7** |

*Table 5.* Ablation study on CPO and TACA modules.

| Category | CPO | TACA | AUC-P ↑ | AP-P ↑ | F1-P ↑ |
|---|---|---|---|---|---|
| capsule | ✓ | ✗ | 98.9 | 71.6 | 65.7 |
| | ✗ | ✓ | 98.8 | 57.2 | 59.8 |
| | ✓ | ✓ | **99.7** | **75.6** | **69.0** |
| grid | ✓ | ✗ | 98.5 | 59.1 | 56.5 |
| | ✗ | ✓ | 98.3 | 52.9 | 54.6 |
| | ✓ | ✓ | **99.0** | **60.1** | **57.2** |
| wood | ✓ | ✗ | 99.2 | 91.8 | 74.7 |
| | ✗ | ✓ | 97.5 | 89.8 | 71.1 |
| | ✓ | ✓ | **99.9** | **94.7** | **76.8** |

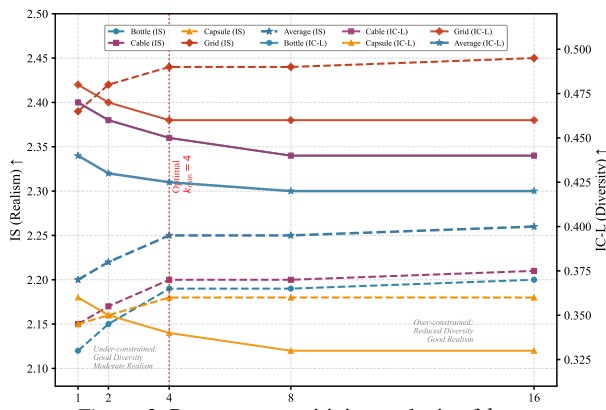

*Figure 3.* Parameter sensitivity analysis of $k_{\min}$.

the-art results on MVTec AD. Our method demonstrates substantial AP improvements over the strongest supervised baseline in challenging categories: 13.4% on capsule, 22.0% on leather, 16.1% on zipper and 12.1% on wood, demonstrating our framework's capacity to generate highly discriminative features that enable precise anomaly localization.

### 5.6. Ablation Study

As shown in Tab. 5, our ablation study validates the complementary roles of CPO and TACA modules. CPO proves crucial for enhancing semantic fidelity through implicit preference alignment, as its removal causes significant performance drops in structurally sensitive categories like capsule and grid. Meanwhile, TACA consistently contributes to maintaining structural diversity across all categories, with its absence leading to noticeable performance degradation. These findings confirm that our framework successfully coordinates optimization objectives to navigate the fidelity-diversity trade-off in anomaly generation.

### 5.7. Parameter Sensitivity Analysis

As illustrated in Fig. 3, we systematically evaluate the impact of $k_{\min} \in \{1, 2, 4, 8, 16\}$ of TACA on generation performance while keeping $k_{\max} = 32$ fixed as in prior work (Jin et al., 2025) due to its effectiveness. Our time-aware capacity allocation exhibits a smooth yet consistent trade-off between realism (IS, dashed) and diversity (IC-L, solid) across categories. The optimal balance is achieved at $k_{\min} = 4$, where insufficient constraints ($k_{\min} < 4$) impair structural fidelity despite preserving diversity, while excessive constraints ($k_{\min} > 4$) reduce diversity without commensurate gains in realism. This behavior systematically confirms that our dynamic rank scheduling effectively regulates the realism–diversity trade-off in few-shot anomaly generation.

# 6. Conclusion

In this paper, we introduced APO, a novel framework for few-shot anomaly generation. Our method tackles preference alignment without human annotation by leveraging anomalies to guide the generative policy through direct preference optimization. To balance diversity and fidelity, we design a Time-Aware Capacity Allocation (TACA) mechanism and a hierarchical sampling strategy for flexible inference. Extensive experiments show that APO synthesizes more realistic and diverse anomalies, significantly boosting detection performance on industrial benchmarks.

# Acknowledgments

This work was supported by the National Natural Science Foundation of China (Grant No. 62476133) and the Fundamental Research Funds for the Central Universities (Grant No. 11300-312200502507).

# Impact Statement

This paper presents work whose goal is to advance the field of Machine Learning, specifically focusing on data-efficient anomaly synthesis. The primary application of our method is to improve the robustness and reliability of automated inspection systems in industrial manufacturing and medical imaging. By reducing the dependency on extensive anomalous data, our work contributes to enhanced quality control standards and operational efficiency. We do not foresee any specific negative ethical aspects or societal consequences associated with this research.

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

## A. Ablation Study on Regularization Coefficient $\beta$

The regularization coefficient $\beta$ governs the fundamental trade-off in our constrained optimization framework, determining the extent to which the model adapts to few-shot anomalies while preventing overfitting. As shown in Tab. 6, $\beta$ exhibits a pronounced effect on generation fidelity and downstream detection performance.

We observe that $\beta = 1000$ achieves optimal balance, yielding superior results in both anomaly realism (IS: 2.14) and detection capability (AP-P: 87.7%). This setting allows sufficient adaptation to learn target anomaly patterns while maintaining the stability essential for few-shot learning. The performance degradation at $\beta = 500$ confirms the risk of overfitting in data-scarce regimes, where excessive adaptation compromises the model's ability to generate structurally coherent anomalies. Conversely, the constrained adaptation at $\beta = 2000$ demonstrates the importance of allowing adequate distributional shift for meaningful anomaly learning. These findings validate that our direct preference optimization provides effective control over the adaptation degree, with $\beta = 1000$ establishing the optimal operating point that enables precise anomaly alignment while preserving the essential characteristics of the base generative model.

*Table 6.* Ablation study on regularization coefficient $\beta$. Results report average performance across MVTec AD categories.

| $\beta$ | IS ↑ | IC-LPIPS ↑ | AUC-P ↑ | AP-P ↑ |
|---|---|---|---|---|
| 500 | 2.08 | 0.39 | 99.4 | 86.2 |
| 1000 | **2.14** | **0.41** | **99.6** | **87.7** |
| 2000 | 2.11 | 0.40 | 99.5 | 87.1 |
| **Best Baseline** | 2.02 | 0.38 | 99.1 | 84.5 |

## B. Additional Theoretical Analysis of Anomaly Preference Optimization

This appendix provides the complete derivation of the key formulas in the main text to ensure theoretical rigor and readability. We begin with the fundamental theory of diffusion models and progressively derive the APO objective function.

### B.1. Trajectory-Level KL Divergence Formulation

We first establish the theoretical foundation at the trajectory level. Let $q(\mathbf{z}_{0:T}|\mathbf{x})$ denote the forward diffusion process and $p_\theta(\mathbf{z}_{0:T}|\mathbf{c})$ the reverse generation process. The KL deviation function defined in the main text is:

$$
\begin{aligned}
\delta(p_\theta, p_{\text{ref}}; \mathbf{x}, \mathbf{c}) = {} & D_{\text{KL}}(q(\mathbf{z}_{0:T}|\mathbf{x})\|p_{\text{ref}}(\mathbf{z}_{0:T}|\mathbf{c})) \\
& - D_{\text{KL}}(q(\mathbf{z}_{0:T}|\mathbf{x})\|p_\theta(\mathbf{z}_{0:T}|\mathbf{c})).
\end{aligned}
\tag{21}
$$

This trajectory-level KL divergence decomposes into per-timestep components:

$$
\begin{aligned}
\delta(p_\theta, p_{\text{ref}}; \mathbf{x}, \mathbf{c}) = \sum_{t=1}^{T} \mathbb{E}_{q(\mathbf{z}_t|\mathbf{x})} & \\
\Big[ D_{\text{KL}}(q(\mathbf{z}_{t-1}|\mathbf{z}_t, \mathbf{x})\|p_{\text{ref}}(\mathbf{z}_{t-1}|\mathbf{z}_t, \mathbf{c})) & \\
- D_{\text{KL}}(q(\mathbf{z}_{t-1}|\mathbf{z}_t, \mathbf{x})\|p_\theta(\mathbf{z}_{t-1}|\mathbf{z}_t, \mathbf{c})) \Big]. &
\end{aligned}
\tag{22}
$$

For each timestep $t$, all three conditionals $q(\mathbf{z}_{t-1}|\mathbf{z}_t, \mathbf{x})$, $p_{\text{ref}}(\mathbf{z}_{t-1}|\mathbf{z}_t, \mathbf{c})$ and $p_\theta(\mathbf{z}_{t-1}|\mathbf{z}_t, \mathbf{c})$ are Gaussians with the *same* covariance but different means. Under the noise-prediction parameterization, we have

$$
q(\mathbf{z}_{t-1}|\mathbf{z}_t, \mathbf{x}) = \mathcal{N}\big(\mathbf{z}_{t-1}; \boldsymbol{\mu}_q(\mathbf{z}_t, \mathbf{z}_0), \sigma_t^2 \mathbf{I}\big),
\tag{23}
$$

$$
p_{\text{ref}}(\mathbf{z}_{t-1}|\mathbf{z}_t, \mathbf{c}) = \mathcal{N}\big(\mathbf{z}_{t-1}; \boldsymbol{\mu}_{\text{ref}}(\mathbf{z}_t, \mathbf{c}, t), \sigma_t^2 \mathbf{I}\big),
\tag{24}
$$

$$
p_\theta(\mathbf{z}_{t-1}|\mathbf{z}_t, \mathbf{c}) = \mathcal{N}\big(\mathbf{z}_{t-1}; \boldsymbol{\mu}_\theta(\mathbf{z}_t, \mathbf{c}, t), \sigma_t^2 \mathbf{I}\big),
\tag{25}
$$

where the variance $\sigma_t^2\mathbf{I}$ is shared, and the means are given by the standard DDPM parameterization:

$$\boldsymbol{\mu}_q(\mathbf{z}_t, \mathbf{z}_0) = \frac{1}{\sqrt{\alpha_t}}\left(\mathbf{z}_t - \frac{1-\alpha_t}{\sqrt{1-\bar{\alpha}_t}}\boldsymbol{\epsilon}\right), \tag{26}$$

$$\boldsymbol{\mu}_{\text{ref}}(\mathbf{z}_t, \mathbf{c}, t) = \frac{1}{\sqrt{\alpha_t}}\left(\mathbf{z}_t - \frac{1-\alpha_t}{\sqrt{1-\bar{\alpha}_t}}\epsilon_{\text{ref}}(\mathbf{z}_t, \mathbf{c}, t)\right), \tag{27}$$

$$\boldsymbol{\mu}_\theta(\mathbf{z}_t, \mathbf{c}, t) = \frac{1}{\sqrt{\alpha_t}}\left(\mathbf{z}_t - \frac{1-\alpha_t}{\sqrt{1-\bar{\alpha}_t}}\epsilon_\theta(\mathbf{z}_t, \mathbf{c}, t)\right), \tag{28}$$

with $\mathbf{z}_t = \sqrt{\bar{\alpha}_t}\mathbf{z}_0 + \sqrt{1-\bar{\alpha}_t}\boldsymbol{\epsilon}$ and $\boldsymbol{\epsilon} \sim \mathcal{N}(\mathbf{0}, \mathbf{I})$.

Since the covariances are identical, the KL divergence between $q$ and a generic Gaussian $p = \mathcal{N}(\boldsymbol{\mu}_p, \sigma_t^2\mathbf{I})$ simplifies to:

$$D_{\text{KL}}(q\|p) = \frac{1}{2\sigma_t^2}\|\boldsymbol{\mu}_q - \boldsymbol{\mu}_p\|_2^2 + \text{const.} \tag{29}$$

Applying this to $p_{\text{ref}}$ and $p_\theta$ and taking the difference gives:

$$\begin{aligned} &D_{\text{KL}}(q(\mathbf{z}_{t-1}|\mathbf{z}_t, \mathbf{x})\|p_{\text{ref}}(\mathbf{z}_{t-1}|\mathbf{z}_t, \mathbf{c})) \\ &\quad - D_{\text{KL}}(q(\mathbf{z}_{t-1}|\mathbf{z}_t, \mathbf{x})\|p_\theta(\mathbf{z}_{t-1}|\mathbf{z}_t, \mathbf{c})) \\ &= \frac{1}{2\sigma_t^2}\left(\|\boldsymbol{\mu}_q - \boldsymbol{\mu}_{\text{ref}}\|_2^2 - \|\boldsymbol{\mu}_q - \boldsymbol{\mu}_\theta\|_2^2\right). \end{aligned} \tag{30}$$

Using the explicit forms of the means, we observe that:

$$\boldsymbol{\mu}_q - \boldsymbol{\mu}_{\text{ref}} = \frac{1-\alpha_t}{\sqrt{\alpha_t(1-\bar{\alpha}_t)}}\left(\epsilon_{\text{ref}}(\mathbf{z}_t, \mathbf{c}, t) - \boldsymbol{\epsilon}\right), \tag{31}$$

$$\boldsymbol{\mu}_q - \boldsymbol{\mu}_\theta = \frac{1-\alpha_t}{\sqrt{\alpha_t(1-\bar{\alpha}_t)}}\left(\epsilon_\theta(\mathbf{z}_t, \mathbf{c}, t) - \boldsymbol{\epsilon}\right), \tag{32}$$

so that:

$$\|\boldsymbol{\mu}_q - \boldsymbol{\mu}_{\text{ref}}\|_2^2 = \frac{(1-\alpha_t)^2}{\alpha_t(1-\bar{\alpha}_t)}\|\epsilon_{\text{ref}}(\mathbf{z}_t, \mathbf{c}, t) - \boldsymbol{\epsilon}\|_2^2, \tag{33}$$

$$\|\boldsymbol{\mu}_q - \boldsymbol{\mu}_\theta\|_2^2 = \frac{(1-\alpha_t)^2}{\alpha_t(1-\bar{\alpha}_t)}\|\epsilon_\theta(\mathbf{z}_t, \mathbf{c}, t) - \boldsymbol{\epsilon}\|_2^2. \tag{34}$$

Substituting these into Eqn. (30) yields:

$$\begin{aligned} &D_{\text{KL}}(q\|p_{\text{ref}}) - D_{\text{KL}}(q\|p_\theta) \\ &= \frac{(1-\alpha_t)^2}{2\sigma_t^2\alpha_t(1-\bar{\alpha}_t)}\left(\|\boldsymbol{\epsilon} - \epsilon_{\text{ref}}(\mathbf{z}_t, \mathbf{c}, t)\|_2^2 - \|\boldsymbol{\epsilon} - \epsilon_\theta(\mathbf{z}_t, \mathbf{c}, t)\|_2^2\right). \end{aligned} \tag{35}$$

Defining:

$$\lambda_t' = \frac{(1-\alpha_t)^2}{\sigma_t^2\alpha_t(1-\bar{\alpha}_t)} \tag{36}$$

as in the main text, and using the standard reparameterization $\mathbf{z}_t = \sqrt{\bar{\alpha}_t}\mathbf{z}_0 + \sqrt{1-\bar{\alpha}_t}\boldsymbol{\epsilon}$ with $t \sim \mathcal{U}(1, T)$ and $\boldsymbol{\epsilon} \sim \mathcal{N}(\mathbf{0}, \mathbf{I})$, the trajectory-level deviation function becomes:

$$\begin{aligned} \delta(p_\theta, p_{\text{ref}}; \mathbf{x}, \mathbf{c}) &= \sum_{t=1}^{T}\mathbb{E}_{q(\mathbf{z}_t|\mathbf{x})}\left[D_{\text{KL}}(q\|p_{\text{ref}}) - D_{\text{KL}}(q\|p_\theta)\right] \\ &= \frac{1}{2}\mathbb{E}_{t\sim\mathcal{U}(0,T),\,\boldsymbol{\epsilon}\sim\mathcal{N}(\mathbf{0},\mathbf{I})}\left[\lambda_t'\left(\|\boldsymbol{\epsilon} - \epsilon_{\text{ref}}(\mathbf{z}_t, \mathbf{c}, t)\|_2^2\right.\right. \\ &\quad\left.\left. - \|\boldsymbol{\epsilon} - \epsilon_\theta(\mathbf{z}_t, \mathbf{c}, t)\|_2^2\right)\right], \end{aligned} \tag{37}$$

which is the tractable deviation function used in the main text.

### B.2. Reward Reparameterization and Preference Modeling

In order to relate the implicit reward to the KL deviation, we extend the optimal policy form to the trajectory level. Analogous to the endpoint case, we write:

$$p_\theta(\mathbf{z}_{0:T}|\mathbf{c}) = \frac{1}{Z(\mathbf{c})}p_{\text{ref}}(\mathbf{z}_{0:T}|\mathbf{c})\exp\left(\frac{\mathcal{R}(\mathbf{z}_{0:T},\mathbf{c})}{\beta}\right), \tag{38}$$

where $Z(\mathbf{c})$ is the partition function that does not depend on the particular trajectory $\mathbf{z}_{0:T}$. Taking logarithms on both sides yields the reward reparameterization:

$$\mathcal{R}(\mathbf{z}_{0:T},\mathbf{c}) = \beta\Big(\log p_\theta(\mathbf{z}_{0:T}|\mathbf{c}) - \log p_{\text{ref}}(\mathbf{z}_{0:T}|\mathbf{c})\Big) + \beta\log Z(\mathbf{c}). \tag{39}$$

We now take expectation with respect to the forward process $q(\mathbf{z}_{0:T}|\mathbf{x})$:

$$\mathbb{E}_{q(\mathbf{z}_{0:T}|\mathbf{x})}\big[\mathcal{R}(\mathbf{z}_{0:T},\mathbf{c})\big] = \beta\Big(\mathbb{E}_{q(\mathbf{z}_{0:T}|\mathbf{x})}[\log p_\theta(\mathbf{z}_{0:T}|\mathbf{c})]$$
$$- \mathbb{E}_{q(\mathbf{z}_{0:T}|\mathbf{x})}[\log p_{\text{ref}}(\mathbf{z}_{0:T}|\mathbf{c})]\Big)$$
$$+ \beta\log Z(\mathbf{c}). \tag{40}$$

On the other hand, by definition of the KL divergence we have:

$$D_{\text{KL}}(q(\mathbf{z}_{0:T}|\mathbf{x})\|p_{\text{ref}}(\mathbf{z}_{0:T}|\mathbf{c})) = \mathbb{E}_{q(\mathbf{z}_{0:T}|\mathbf{x})}\big[\log q(\mathbf{z}_{0:T}|\mathbf{x})$$
$$- \log p_{\text{ref}}(\mathbf{z}_{0:T}|\mathbf{c})\big], \tag{41}$$

$$D_{\text{KL}}(q(\mathbf{z}_{0:T}|\mathbf{x})\|p_\theta(\mathbf{z}_{0:T}|\mathbf{c})) = \mathbb{E}_{q(\mathbf{z}_{0:T}|\mathbf{x})}\big[\log q(\mathbf{z}_{0:T}|\mathbf{x})$$
$$- \log p_\theta(\mathbf{z}_{0:T}|\mathbf{c})\big]. \tag{42}$$

Subtracting these two expressions, we obtain:

$$D_{\text{KL}}(q(\mathbf{z}_{0:T}|\mathbf{x})\|p_{\text{ref}}(\mathbf{z}_{0:T}|\mathbf{c})) - D_{\text{KL}}(q(\mathbf{z}_{0:T}|\mathbf{x})\|p_\theta(\mathbf{z}_{0:T}|\mathbf{c}))$$
$$= \mathbb{E}_{q(\mathbf{z}_{0:T}|\mathbf{x})}\big[\log p_\theta(\mathbf{z}_{0:T}|\mathbf{c}) - \log p_{\text{ref}}(\mathbf{z}_{0:T}|\mathbf{c})\big]. \tag{43}$$

Using the deviation function defined in the main text,

$$\delta(p_\theta, p_{\text{ref}};\mathbf{x},\mathbf{c}) = D_{\text{KL}}(q(\mathbf{z}_{0:T}|\mathbf{x})\|p_{\text{ref}}(\mathbf{z}_{0:T}|\mathbf{c}))$$
$$- D_{\text{KL}}(q(\mathbf{z}_{0:T}|\mathbf{x})\|p_\theta(\mathbf{z}_{0:T}|\mathbf{c})), \tag{44}$$

we can rewrite the expectation difference in Eqn. (40) as:

$$\mathbb{E}_{q(\mathbf{z}_{0:T}|\mathbf{x})}[\log p_\theta(\mathbf{z}_{0:T}|\mathbf{c})] - \mathbb{E}_{q(\mathbf{z}_{0:T}|\mathbf{x})}[\log p_{\text{ref}}(\mathbf{z}_{0:T}|\mathbf{c})]$$
$$= \delta(p_\theta, p_{\text{ref}};\mathbf{x},\mathbf{c}). \tag{45}$$

Substituting this into Eqn. (40) gives the desired relation:

$$\mathbb{E}_{q(\mathbf{z}_{0:T}|\mathbf{x})}[\mathcal{R}(\mathbf{z}_{0:T},\mathbf{c})] = \beta\cdot\delta(p_\theta, p_{\text{ref}};\mathbf{x},\mathbf{c}) + \beta\log Z(\mathbf{c}), \tag{46}$$

which is the equation stated in the main text. It shows that, up to an additive constant independent of the particular anomaly sample $\mathbf{x}$, maximizing the expected implicit reward is equivalent to maximizing the KL deviation $\delta(p_\theta, p_{\text{ref}};\mathbf{x},\mathbf{c})$.

To construct a practical optimization objective, we now connect the trajectory-level deviation $\delta(p_\theta, p_{\text{ref}};\mathbf{x},\mathbf{c})$ with an instantaneous deviation in noise space. From the previous subsection, we have already obtained the closed-form expression

$$\delta(p_\theta, p_{\text{ref}};\mathbf{x},\mathbf{c}) = \frac{1}{2}\mathbb{E}_{t\sim\mathcal{U}(1,T),\,\boldsymbol{\epsilon}\sim\mathcal{N}(\mathbf{0},\mathbf{I})}$$
$$\big[\lambda'_t\big(\|\boldsymbol{\epsilon} - \boldsymbol{\epsilon}_{\text{ref}}(\mathbf{z}_t,\mathbf{c},t)\|_2^2 - \|\boldsymbol{\epsilon} - \boldsymbol{\epsilon}_\theta(\mathbf{z}_t,\mathbf{c},t)\|_2^2\big)\big], \tag{47}$$

where $\lambda'_t = \frac{(1-\alpha_t)^2}{\sigma_t^2 \alpha_t (1-\bar{\alpha}_t)}$ and $\mathbf{z}_t = \sqrt{\bar{\alpha}_t} \mathbf{z}_0 + \sqrt{1-\bar{\alpha}_t} \boldsymbol{\epsilon}$.

In the main text, the instantaneous deviation is defined as

$$\Delta = \|\epsilon_\theta(\mathbf{z}_t, \mathbf{c}, t) - \boldsymbol{\epsilon}\|_2^2 - \|\epsilon_{\text{ref}}(\mathbf{z}_t, \mathbf{c}, t) - \boldsymbol{\epsilon}\|_2^2, \tag{48}$$

which is simply the negative of the bracketed term above, since squared distances are symmetric:

$$\|\boldsymbol{\epsilon} - \epsilon_{\text{ref}}\|_2^2 - \|\boldsymbol{\epsilon} - \epsilon_\theta\|_2^2 = \|\epsilon_{\text{ref}} - \boldsymbol{\epsilon}\|_2^2 - \|\epsilon_\theta - \boldsymbol{\epsilon}\|_2^2$$
$$= -\left( \|\epsilon_\theta - \boldsymbol{\epsilon}\|_2^2 - \|\epsilon_{\text{ref}} - \boldsymbol{\epsilon}\|_2^2 \right) = -\Delta. \tag{49}$$

Substituting this identity into the deviation expression yields the precise relation between the trajectory-level deviation and the instantaneous deviation:

$$\delta(p_\theta, p_{\text{ref}}; \mathbf{x}, \mathbf{c}) = -\frac{1}{2} \mathbb{E}_{t\sim\mathcal{U}(1,T),\, \boldsymbol{\epsilon}\sim\mathcal{N}(\mathbf{0},\mathbf{I})} \left[ \lambda'_t \Delta \right]. \tag{50}$$

Combining Eqn. (50) with the reward reparameterization:

$$\mathbb{E}_{q(\mathbf{z}_{0:T}|\mathbf{x})} \left[ \mathcal{R}(\mathbf{z}_{0:T}, \mathbf{c}) \right] = \beta \cdot \delta(p_\theta, p_{\text{ref}}; \mathbf{x}, \mathbf{c}) + \beta \log Z(\mathbf{c}), \tag{51}$$

we obtain:

$$\mathbb{E}_{q(\mathbf{z}_{0:T}|\mathbf{x})} \left[ \mathcal{R}(\mathbf{z}_{0:T}, \mathbf{c}) \right] = -\frac{\beta}{2} \mathbb{E}_{t,\boldsymbol{\epsilon}} \left[ \lambda'_t \Delta \right] + \beta \log Z(\mathbf{c}). \tag{52}$$

Defining a time-dependent scaling factor:

$$\beta_t = \frac{1}{2} \beta \lambda'_t, \tag{53}$$

this can be rewritten as:

$$\mathbb{E}_{q(\mathbf{z}_{0:T}|\mathbf{x})} \left[ \mathcal{R}(\mathbf{z}_{0:T}, \mathbf{c}) \right] = -\mathbb{E}_{t,\boldsymbol{\epsilon}} \left[ \beta_t \Delta \right] + \beta \log Z(\mathbf{c}). \tag{54}$$

The constant term $\log Z(\mathbf{c})$ does not affect optimization with respect to $\theta$, so the effective reward margin is proportional to $-\beta_t \Delta$. Intuitively, when the policy denoiser $\epsilon_\theta$ is closer to the true noise than the reference $\epsilon_{\text{ref}}$, we have $\Delta < 0$ and thus $-\beta_t \Delta > 0$, indicating a higher implicit reward for the policy.

To convert this margin into a practical learning objective, we adopt the Bradley–Terry preference model. For each noisy latent state $(\mathbf{z}_t, \mathbf{c}, t, \boldsymbol{\epsilon})$, we model the probability that the policy $p_\theta$ is preferred over the reference model $p_{\text{ref}}$ as

$$P\left( p_\theta \succ p_{\text{ref}} \mid \mathbf{x}, \mathbf{c}, t, \boldsymbol{\epsilon} \right) = \text{sigmoid}\left( -\beta_t \Delta \right), \tag{55}$$

where $\text{sigmoid}(\cdot)$ is the sigmoid function and $-\beta_t \Delta$ serves as the preference logit. Under this formulation, a better policy (with smaller denoising error than the reference, i.e., $\Delta < 0$) yields a positive logit and thus a preference probability greater than $0.5$.

Maximizing this preference probability over the training distribution is equivalent to minimizing the negative log-likelihood

$$\mathcal{L}_{\text{APO}} = \mathbb{E}_{t\sim\mathcal{U}(0,T),\, \boldsymbol{\epsilon}\sim\mathcal{N}(\mathbf{0},\mathbf{I})} \left[ -\log \text{sigmoid}\left( -\beta_t \Delta \right) \right], \tag{56}$$

which is exactly the APO objective used in the main text. This derivation shows that the instantaneous deviation $\Delta$ provides a consistent bridge from the trajectory-level KL deviation to a DPO-style preference loss, while the time-dependent factor $\beta_t$ reflects the contribution of each diffusion timestep as induced by the ELBO weights $\lambda'_t$.

### B.3. Final APO Objective

From the previous subsection, the trajectory-level deviation can be written in latent noise space as:

$$\delta(p_\theta, p_{\text{ref}}; \mathbf{x}, \mathbf{c}) = \frac{1}{2} \mathbb{E}_{t\sim\mathcal{U}(0,T),\, \boldsymbol{\epsilon}\sim\mathcal{N}(\mathbf{0},\mathbf{I})}$$
$$\left[ \lambda'_t \left( \|\epsilon_\theta(\mathbf{z}_t, \mathbf{c}, t) - \boldsymbol{\epsilon}\|_2^2 - \|\epsilon_{\text{ref}}(\mathbf{z}_t, \mathbf{c}, t) - \boldsymbol{\epsilon}\|_2^2 \right) \right], \tag{57}$$

which matches Eqn. (14) in the main text. For notational convenience, we define the instantaneous deviation:

$$\Delta = \|\epsilon_\theta(\mathbf{z}_t, \mathbf{c}, t) - \boldsymbol{\epsilon}\|_2^2 - \|\epsilon_{\text{ref}}(\mathbf{z}_t, \mathbf{c}, t) - \boldsymbol{\epsilon}\|_2^2, \tag{58}$$

so that:

$$\delta(p_\theta, p_{\text{ref}}; \mathbf{x}, \mathbf{c}) = \frac{1}{2}\mathbb{E}_{t,\epsilon}\big[\lambda_t'\Delta\big]. \tag{59}$$

Combining this expression with the reward reparameterization,

$$\mathbb{E}_{q(\mathbf{z}_{0:T}|\mathbf{x})}\big[\mathcal{R}(\mathbf{z}_{0:T}, \mathbf{c})\big] = \beta \cdot \delta(p_\theta, p_{\text{ref}}; \mathbf{x}, \mathbf{c}) + \beta \log Z(\mathbf{c}), \tag{60}$$

we obtain:

$$\mathbb{E}_{q(\mathbf{z}_{0:T}|\mathbf{x})}\big[\mathcal{R}(\mathbf{z}_{0:T}, \mathbf{c})\big] = \frac{\beta}{2}\mathbb{E}_{t,\epsilon}\big[\lambda_t'\Delta\big] + \beta \log Z(\mathbf{c}). \tag{61}$$

For a fixed tuple $(\mathbf{x}, \mathbf{c}, t, \boldsymbol{\epsilon})$, this suggests a local reward margin:

$$m(\mathbf{x}, \mathbf{c}, t, \boldsymbol{\epsilon}) = \frac{\beta}{2}\lambda_t'\Delta. \tag{62}$$

To match the notation in the main text, we reparameterize this margin via a time-dependent weight:

$$\beta_t := -\frac{1}{2}\beta\lambda_t', \tag{63}$$

so that:

$$m(\mathbf{x}, \mathbf{c}, t, \boldsymbol{\epsilon}) = -\beta_t\Delta. \tag{64}$$

In other words, the effective reward margin is proportional to the scaled deviation $-\beta_t\Delta$ used in Eqn. (15).

Adopting the Bradley–Terry model at the denoising level, we model the probability that the policy $p_\theta$ is preferred over the reference model $p_{\text{ref}}$ as:

$$P\big(p_\theta \succ p_{\text{ref}} \mid \mathbf{x}, \mathbf{c}, t, \boldsymbol{\epsilon}\big) = \text{sigmoid}\big(-\beta_t\Delta\big), \tag{65}$$

where the logit is exactly the margin $m(\mathbf{x}, \mathbf{c}, t, \boldsymbol{\epsilon}) = -\beta_t\Delta$. Maximizing this preference probability over the training distribution is equivalent to minimizing the negative log-likelihood:

$$\begin{aligned}
\mathcal{L}_{\text{APO}} &= \mathbb{E}_{t\sim\mathcal{U}(0,T),\, \epsilon\sim\mathcal{N}(\mathbf{0},\mathbf{I})}\Big[-\log\text{sigmoid}\big(-\beta_t\Delta\big)\Big] \\
&= \mathbb{E}_{t,\epsilon}\Big[-\log\text{sigmoid}\big(-\beta_t\big(\|\epsilon_\theta(\mathbf{z}_t, \mathbf{c}, t) - \boldsymbol{\epsilon}\|_2^2 \\
&\quad - \|\epsilon_{\text{ref}}(\mathbf{z}_t, \mathbf{c}, t) - \boldsymbol{\epsilon}\|_2^2\big)\big)\Big],
\end{aligned} \tag{66}$$

which recovers the APO objective in Eqn. (15) of the main text. This final expression is mathematically consistent with the trajectory-level KL deviation, the reward reparameterization, and the time-dependent weighting scheme, thereby providing a rigorous foundation for the proposed APO model.

