# OpenReview forum: "Anomaly-Preference Image Generation"
_ICML.cc/2026/Conference — ICML 2026 regular_

### Official Review · Reviewer_pCGf · 2026-03-02

**Soundness:** 3
**Presentation:** 3
**Significance:** 2
**Originality:** 2
**Overall Recommendation:** 4
**Confidence:** 3

**Summary:**

To improve the quality of the synthesized anomaly images, this paper proposes the Constrained Preference Optimization. Moreover, a Time-Aware Capacity Allocation module is introduced to better learn high-noise structural features and low-noise fine-grained fidelity representations. During inference, a hierarchical sampling strategy is employed to further strength the generation capability. Experiments show the superior performance of the proposed method.

**Compliance With Llm Reviewing Policy:**

Affirmed.

**Final Justification:**

I acknowledge the authors' efforts in advancing anomaly image generation. Although related techniques have been studied in the broader context of diffusion-based image generation, this work offers a more focused exploration in the anomaly domain. I have therefore increased my score to reflect this aspect.

**Key Questions For Authors:**

1. The ablation includes sensitivity analysis of hyper-parameter $k_{min}$, but the corresponding ablation of $k_{max}$ is missing.
2. A more detailed discuss of the weights of $s_{align}$ would clarify the contribution.
3. The Datasets part indicates that there are approximately 200 images per industrial category. However, few-shot learning typically refers to scenarios with only a small number of training samples. Can the authors clarify this?
4. Could the authors provide more details on how these baseline methods were trained?
5. When comparing with the unsupervised methods, could the author clarify the rationale for including them as baselines, given that the proposed method does not operate under an unsupervised setting?

**Limitations:**

No limitation is discussed. The limitations of real-world deployment could be further discussed.

**Strengths And Weaknesses:**

Strengths: The proposed Constrained Preference Optimization and the Time-Aware Capacity Allocation are technically sound. Sufficient empirical evidence and supporting analysis are provided in the supplementary. The paper is easy to follow with straightforward performance gains. The evaluation metrics are well organized, validating the performance of their proposed method.

Weakness: Though I am not an expert in anomaly image generation. The proposed Constrained Preference Optimization appears much conceptually similar to DPO, particularly given that it directly removes the negative guidance part. The authors claim that the CPO improves the few-shot adaptation ability of DPO, but the provided evidence does not clearly establish this connection. Moreover, attributing structural/texture features to different diffusion timesteps[1,2] has been extensively studied in the broader diffusion image generation literature, which makes the technique's contribution less significant. However, maybe this topic is less explored in Anomaly Image Generation, and I will consider the advice of other reviewers to decide.

[1] Zhang Y, Dong W, Tang F, et al. Prospect: Prompt spectrum for attribute-aware personalization of diffusion models. (TOG 2023)

[2] Agarwal A, Karanam S, Shukla T, et al. An image is worth multiple words: Multi-attribute inversion for constrained text-to-image synthesis (WACV 2025)

---

> ### Author Rebuttal · Authors · 2026-03-28
>
> Thanks for your insightful and constructive comments.
>
> **Reponse to Weaknesses:**
>
> While sharing a conceptual lineage with DPO, CPO fundamentally reformulates the preference paradigm. Standard DPO requires explicit pairwise samples, but defining a universal "rejected" industrial defect is ill-posed. Instead of arbitrarily removing a negative bound, CPO shifts preference from comparing generated samples to comparing the denoising trajectories of two models. By optimizing a KL constraint against a reference model, CPO directly aligns the policy with the target distribution.
>
> Furthermore, while timestep-dependent priors are prevalent in general diffusion literature [1, 2], our integration is fundamentally different. Methods like [1, 2] use these priors primarily for prompt manipulation or latent inversion. In contrast, our TACA module embeds this prior into the optimization objective by dynamically regulating the network's trainable capacity. This structurally forces the policy to preserve normal backgrounds at high-noise stages while dedicating capacity to fine defect textures at low-noise stages. Integrating temporal heuristics as a dynamic capacity constraint directly into the preference learning process is highly novel in the field of few-shot anomaly image generation.
>
> **Reponse to Key Questions:**
>
> 2. Ablation Studies on $k_{max}$ and $s_{align}$. We have included the requested analyses to clarify these hyper-parameters (evaluated on MVTec AD). Ablation on $k_{max}$ with $k_{min}=4$: While $k_{min}$ preserves structure, $k_{max}$ dictates the maximum capacity for fine-grained texture synthesis at low-noise steps $t \to 0$.
> | $k_{max}$ |IC-LPIPS|AUC-P (%)|AP-P (%)|F1-P (%) |
> |----|----|------|----|-----|
> |16|0.40|96.2|84.3|78.5|
> |32|**0.41** |**99.6**|**87.7**|**81.6**|
> |64|0.38|98.8|85.5|79.3|
>
>    Restricting $k_{max}$ limits texture fidelity, while excessive capacity ($k_{max}=64$) causes overfitting. $k_{max}=32$ provides the optimal balance.Analysis of $s_{align}$: This weight controls the injection intensity of the optimized anomaly deviation ($\Delta_{align} = \epsilon_\theta - \epsilon_{ref}$).
>
>    |$s_{align}$| IC-LPIPS | AUC-P (%) | AP-P (%) | F1-P (%) |
>    |----|-----|-----|-----|-----|
>    |1.0|0.36|98.5|82.1|76.4|
>    |2.0|0.39|99.1|85.4|79.2|
>    |3.0|**0.41**|**99.6**|**87.7**|**81.6**|
>    |4.0|0.38|98.5|84.2|77.5|
>
>    Smaller weights ($\le 2.0$) generate indistinct anomalies, while excessive weights ($4.0$) over-amplify deviation and corrupt normal backgrounds. $s_{align}=3.0$ optimally balances anomaly prominence and background preservation.
>
> 3. The “approximately 200 images per category” in the Datasets section refers to the normal training images, not to the anomaly samples used in our few-shot setting. In industrial anomaly detection, normal images are usually abundant and are used to establish the normal data distribution.
>
>    Our use of “few-shot” refers specifically to the anomalous reference samples used for generation. In MVTec AD, many defect types contain only about 15–30 anomaly images in total, and our method uses only a small subset of them (typically about 5 samples per defect type). We will revise the Datasets section to separate the statistics of normal and anomalous samples more clearly.
>
> 4. To ensure a rigorous and fair comparison, we trained all baselines (AnomalyDiffusion, DualAnoDiff, SeaS) using their official source codes and default hyperparameters on the exact same data split as APO, utilizing only one-third of the anomalous images. Subsequently, to objectively evaluate the synthesized data, we trained a standard U-Net from scratch using 1,000 image-mask pairs generated by each respective model. We will detail these specific training configurations in the revised appendix to guarantee full reproducibility.
>
> 5.  We include unsupervised methods because they are the most relevant practical reference when anomaly samples are unavailable. This comparison shows the practical value of our setting: whether introducing only a few anomaly examples can bring clear gains over widely used unsupervised alternatives. We make this rationale clearer in the paper and distinguish these methods from the few-shot baselines more explicitly.

---

> > ### Author Rebuttal · Reviewer_pCGf · 2026-04-03
> >
> > I acknowledge the authors' efforts in advancing anomaly image generation. Although related techniques have been studied in the broader context of diffusion-based image generation, this work offers a more focused exploration in the anomaly domain. I have therefore increased my score to reflect this aspect.

---

> > > ### Author Response · Authors · 2026-04-03
> > >
> > > Dear Reviewer pCGf,
> > >
> > > Thank you very much for your careful reading of our manuscript and for your thoughtful follow-up feedback. We are grateful for the time and effort you spent revisiting our paper, as well as for your consideration of our rebuttal and the additional results we provided.
> > >
> > > Your comments have been very helpful in identifying ways to further improve the manuscript. We will continue refining the paper by carefully addressing your suggestions and by incorporating the broader feedback from all reviewers into the revised version.
> > >
> > > We sincerely appreciate your valuable evaluation and continued consideration of our work.
> > >
> > > Sincerely,
> > >
> > > Authors

---

### Official Review · Reviewer_Jcv8 · 2026-03-04

**Soundness:** 2
**Presentation:** 2
**Significance:** 2
**Originality:** 2
**Overall Recommendation:** 4
**Confidence:** 3

**Summary:**

This paper proposes Anomaly Preference Optimization (APO), a framework for few-shot anomaly image generation that reformulates anomaly synthesis as a preference learning problem within diffusion models. Instead of relying on human-annotated preference pairs, the method derives implicit preference signals from real anomaly samples by comparing denoising trajectory deviations between the current model and a reference diffusion model. In addition, the authors introduce a Time-Aware Capacity Allocation (TACA) mechanism that dynamically allocates model capacity across diffusion timesteps to balance structural diversity and fine-grained fidelity, along with a hierarchical sampling strategy to control anomaly alignment during inference. Experiments on the MVTec benchmark show that the proposed method achieves superior performance in anomaly generation quality and improves downstream anomaly detection and localization performance compared with existing approaches.

Although the authors follow the experimental protocol used in prior work, it is unclear whether using one-third of the original anomaly samples can genuinely be considered a few-shot setting. In many areas such as few-shot classification, the term “few-shot” usually refers to scenarios with only a handful of samples per class. Therefore, using one-third of the dataset may not fully align with the commonly accepted definition. Moreover, it is unclear how much practical benefit reducing the number of anomaly samples by two-thirds brings to real-world anomaly detection applications. This point would benefit from further discussion and clarification.

In addition, recent studies commonly evaluate anomaly generation or detection methods on datasets such as VisA and MPDD. These datasets are not included in the experiments, which makes the empirical validation somewhat limited.

Finally, in the methodology section, it is difficult to clearly identify the main technical innovations and advantages of the proposed model compared with existing approaches. A clearer explanation of the methodological contributions would help improve the readability and impact of the paper.

**Compliance With Llm Reviewing Policy:**

Affirmed.

**Final Justification:**

Score increased by one point

**Key Questions For Authors:**

See summary

**Limitations:**

See summary

**Strengths And Weaknesses:**

See summary

---

> ### Author Rebuttal · Authors · 2026-03-28
>
> Thanks for your insightful and constructive comments.
>
> 1. The one-third protocol was adopted for consistency with prior work, but we agree that a ratio alone does not clearly convey the few-shot regime. In MVTec AD, many defect categories contain only 15–30 anomaly images in total, so using one-third of them typically means only 5–10 anomaly samples per defect type. We will make this absolute sample count explicit in the paper, since it better reflects the actual data regime than the ratio itself.
>
>     To address the stricter interpretation of few-shot more directly, we additionally evaluated K-shot settings with $K \in$ \{1, 3, 5\} . APO remains effective even at K=1 and consistently outperforms AnoDiff at K=3 and K=5, which shows that the method is not limited to the one-third protocol.
>
>    |Method|K-shot|AUC-P|AP-P|F1-P|AUC-I|
>    |------|------|-----|----|----|-----|
>    |AnoDiff|K=1|82.3|31.5|34.2|84.1|
>    |Ours|K=1|**94.2**|**61.4**|**58.6**|**93.5**|
>    |AnoDiff|K=3|89.5|45.8|48.7|91.2|
>    |Ours|K=3|**96.5**|**72.1**|**67.8**|**95.8**|
>    |AnoDiff|K=5|93.1|58.2|60.1|94.6|
>    |Ours|K=5|**97.8**|**79.3**|**73.5**|**97.2**|
>
>    The practical benefit is that the model can be adapted from only a handful of real defect samples, instead of waiting to collect substantially more rare anomaly examples before deployment. We will add both this discussion and the K-shot results to clarify the few-shot definition and its practical value.
>
>
> 2. Following your suggestion, we conducted additional experiments on both datasets. We compared our approach against recent baselines across downstream detection/localization and generation quality (IS, IL).
>
>    **Table 1: Downstream Performance on MPDD**
>    |Method|AUC-P|AP-P|F1-P|AUC-I|AP-I|F1-I|
>    | :--- | :---: | :---: | :---: | :---: | :---: | :---: |
>    |DRAEM|85.2|36.7|38.3|87.6|73.4|67.8|
>    |DFMGAN|92.4|41.6|45.5|92.5|78.5|78.9|
>    |AnoDiff|95.0|47.6|49.5|95.3|88.5|88.0|
>    |Ours|**96.5**|**54.2**|**55.8**|**96.8**|**92.3**|**92.4**|
>
>    **Table 2: Downstream Performance on VisA**
>    |Method|AUC-P|AP-P|F1-P| AUC-I| AP-I|F1-I|
>    | :--- | :---: | :---: | :---: | :---: | :---: | :---: |
>    |DRAEM|88.8|25.4|24.6|84.1| 88.2|87.4|
>    |DFMGAN|95.2|39.9|40.2|87.1| 85.2|85.7|
>    |AnoDiff |98.0|33.2|37.1|96.2|96.9|**92.6**|
>    |Ours|**98.8**|**45.3**|**48.2**|**98.3**|**97.7**|92.5|
>
>    **Table 3: Generation Quality (IS & IL)**
>    |Dataset|Method|IS|IL|
>    | :--- | :--- | :---: | :---: |
>    |MPDD|DFMGAN/AnoDiff |1.31/1.60|0.31/0.43|
>    || Ours|**1.91**|**0.64**|
>    | VisA|DFMGAN/AnoDiff |1.26/1.45|1.12/1.73|
>    || Ours|**1.88**|**2.34**|
>
>    As shown above, our method consistently outperforms the strongest recent baselines (e.g., AnoDiff) across both metrics and datasets. These new evaluations conclusively demonstrate our model's robust generalization to diverse and unsaturated industrial scenarios. We will include these comprehensive results in the revised manuscript.
>
> 3. Compared with existing methods, the main technical contribution of APO is that it formulates few-shot anomaly generation as learning from real anomaly examples through policy/base-model deviation, rather than relying on hand-crafted defect simulation, explicit masking, or global latent editing. This gives the model a direct training signal tied to real defect patterns.
>
>
>    **The method has three main contributions.** First, APO defines anomaly synthesis through the learned deviation between the policy model and the base model on real anomaly trajectories, which provides a principled way to align generation with the reference defects. Second, TACA controls adaptation strength across denoising steps, so the model can capture defect details while preserving the normal object structure. Third, the same learned deviation is reused at inference time as the anomaly guidance signal, and can also be aggregated for zero-shot localization.
>
>
>     **These points are the main methodological differences and advantages of APO:** it learns from real anomaly patterns, preserves normal structure during generation, and uses one unified signal for both anomaly synthesis and localization.

---

> > ### Author Rebuttal · Reviewer_Jcv8 · 2026-04-03
> >
> > Thank you for your reply.

---

> > > ### Author Response · Authors · 2026-04-03
> > >
> > > Dear Reviewer Jcv8,
> > >
> > > We would like to express our sincere gratitude for your careful review and thoughtful comments. We truly appreciate your recognition of our additional experiments and rebuttal efforts, as well as the time and attention you devoted to re-evaluating our manuscript.
> > >
> > > Your insightful and constructive suggestions are extremely valuable to us and have helped us further improve the quality and clarity of our work. We will carefully address your comments and incorporate the relevant revisions into the final manuscript.
> > >
> > > Thank you again for your time, support, and valuable feedback.
> > >
> > > Sincerely,
> > >
> > > Authors

---

### Official Review · Reviewer_5h3t · 2026-03-11

**Soundness:** 3
**Presentation:** 3
**Significance:** 3
**Originality:** 3
**Overall Recommendation:** 4
**Confidence:** 4

**Summary:**

This paper introduces a framework named "anomaly preference optimisation" for generating anomalous samples in few-shot settings. The framework formulates anomaly generation as a preference learning problem using an implicit preference that leverages real anomalies as positive references, and uses the alignment deviation between a policy model and a base model for learning. Additionally, a time-aware capacity allocation module and a hierarchical sampling strategy are proposed to further enhance inference. Experiments on benchmarks demonstrate good performance in generation quality and downstream anomaly detection tasks.

**Compliance With Llm Reviewing Policy:**

Affirmed.

**Final Justification:**

Concerns mostly solved.

**Key Questions For Authors:**

See weaknesses.

**Limitations:**

See weaknesses.

**Strengths And Weaknesses:**

Strengths:
1. The proposed method overall is reasonable.
2. The experiments look good.
3. The writing overall is good.

Weaknesses:

1. The basic motivation and formulation are a bit confusing. By definition, "anomaly" means anything out of the distribution of the normal sample. Therefore, it is hard to find the "target distribution" of "anomaly". Maybe it is better to formulate the problem as an "out-of-distribution" problem, rather than an alignment problem to a target distribution?

2. The method itself is reasonable, but the framing as "preference learning" is confusing. As in Algorithm1 Line 9, the framework is mainly based on "alignment deviation", and there is no actual "preference" in the framework. The authors call it "implicit preference", but it is more like storytelling to me.

3. The method should be discussed with the large body of the literature on "few-shot generation". There are many papers on that topic that do not necessarily generate anomaly samples, but share a similar goal and method.

4. The technique in Sections 4.3 and 4.4 is reasonable, but mainly heuristic. And there are also many existing methods about guidance and inference time enhancement, not necessarily for anomaly, but share a similar goal and method. Should be discussed.

5. Table 4 has been highly saturated and might not be meaningful.

---

> ### Author Rebuttal · Authors · 2026-03-28
>
> Thanks for your insightful and constructive comments.
>
> 1. We agree that, in the broad sense, anomalies are out-of-distribution relative to normal data. However, our setting is not to model arbitrary OOD deviations. In few-shot anomaly generation, we are given a small set of real anomaly examples, and the goal is to synthesize new samples that follow the patterns of these reference defects.
> Therefore, the “target distribution” in our paper does not mean a universal distribution of all anomalies. It refers only to the empirical distribution defined by the provided few-shot anomaly set. In this sense, our objective is not generic OOD generation, but alignment to the specific defect patterns observed in the reference anomalies. We clarify this more explicitly in the revision.
>
> 2. To clarify the point raised here, in APO, “preference” does not refer to human preference or to an explicit pair of generated samples. It refers to a relative comparison between two models on the same real anomaly: if the policy model explains the same anomaly better than the frozen base model, that outcome is treated as preferred.
> Under this definition, the alignment deviation in Algorithm 1 Line 9 is exactly the score that realizes this comparison. So it is not separate from “preference”; it is the quantity used to measure the policy model’s relative advantage over the base model on the same reference anomaly. We  clarify this definition more explicitly in the revision.
>
> 3. We expand the Related Work to discuss the broader literature on few-shot generation, since many of these methods share the same high-level goal of synthesizing new samples from very limited examples. At the same time, we will clarify the key difference in setting: general few-shot generation usually aims to reproduce a class, style, or domain from a few exemplars, whereas APO focuses on few-shot anomaly generation, where the generated content must match the defect patterns in the reference anomalies while preserving the normal object structure. We revise the paper to better position APO with respect to this broader few-shot generation literature.
>
> 4. The heuristic techniques in Sections 4.3 and 4.4 are related to existing work on guidance and inference-time enhancement [1–4], and we discuss these connections more clearly in the revised paper. Section 4.3 builds on the heuristic that early diffusion steps mainly affect overall structure, while later steps mainly affect local details [1,2]. In our method, this is implemented as TACA, which controls adaptation strength across denoising so that the base model’s structural prior is preserved early and defect-specific details are learned later. Section 4.4 builds on the heuristic of separating semantic guidance from anomaly-specific guidance, as in prior guided generation methods [3,4]. In our method, however, the anomaly guidance is not added as a separate inference trick; it is directly defined by the learned deviation between the policy model and the base model, which is also the signal used for anomaly alignment.
>
>
>
> 5. To address the concern that Table 4 may be saturated, we add results on two more challenging benchmarks, MPDD and VisA, which provide a more discriminative evaluation. APO consistently outperforms the recent strong baseline AnoDiff on both datasets at both the pixel and image levels, with clear gains in AP-P / F1-P and competitive or better AUC-I / AP-I. The generation metrics (IS/IL) also improve consistently, indicating better sample quality and diversity. These additional results show that the advantage of APO is not specific to a saturated benchmark and remains visible on harder datasets.
> |Dataset|Method|AUC-P|AP-P|F1-P|AUC-I|AP-I|F1-I|IS|IL|
> |-----|----|----|----|----|-----|----|----|--|--|
> |MPDD|AnoDiff|95.0|47.6|49.5|95.3|88.5|88.0|1.60|0.43|
> |MPDD|Ours|**96.5**|**54.2**|**55.8**|**96.8**|**92.3**|**92.4**|**1.91**|**0.64**|
> |VisA|AnoDiff|98.0|33.2|37.1|96.2|96.9|92.6|1.45|1.73|
> |VisA|Ours|**98.8**|**45.3**|**48.2**|**98.3**|**97.7**|**92.5**|**1.88**|**2.34**|
>
>
>
> [1] Adaptive Non-Uniform Timestep Sampling for Accelerating Diffusion Model Training
>
> [2] Beta-tuned timestep diffusion model
>
> [3] Styledrop: Text-to-image generation in any style
>
> [4] Muse: Text-to-image generation via masked generative transformers

---

> > ### Author Rebuttal · Reviewer_5h3t · 2026-04-03
> >
> > Mostly resolved

---

> > > ### Author Response · Authors · 2026-04-03
> > >
> > > Dear Reviewer 5h3t,
> > >
> > > We sincerely appreciate your continued engagement with our submission and your thoughtful follow-up remarks. Thank you for carefully considering our rebuttal and the newly added experimental results.
> > >
> > > Your comments and suggestions have been highly helpful in guiding us to further strengthen the presentation and overall quality of the paper. We will thoroughly revise the manuscript by taking your feedback into account, along with the recommendations provided by the other reviewers.
> > >
> > > Thank you once again for your valuable time and considerate evaluation.
> > >
> > > Sincerely,
> > >
> > > Authors

---

### Official Review · Reviewer_zqBA · 2026-03-13

**Soundness:** 3
**Presentation:** 2
**Significance:** 2
**Originality:** 2
**Overall Recommendation:** 4
**Confidence:** 2

**Summary:**

This paper focuses on few-shot anomaly generation.  The main issue with current approaches is balancing realism and diversity—they tend to either overfit or suffer from distribution misalignment. To resolve this, the authors introduce APO, which treats anomaly generation as a preference learning problem. The method relies on three main technical components: an implicit preference alignment mechanism, a time-aware capacity allocation module, and a hierarchical sampling strategy.

**Compliance With Llm Reviewing Policy:**

Affirmed.

**Key Questions For Authors:**

1. Could the author explain what heuristics are used in this submission?


2. Is the TACA schedule hardcoded, or does it dynamically adjust based on the specific type of anomaly?

**Limitations:**

see weekness

**Strengths And Weaknesses:**

pos:
1. This submission aims to treat few-shot generation as a preference learning problem. The topic is novel and may help some special scenarios.
2. A relevant question explored by the study is how to align these distributions without relying on expensive human labeling, which they solve cleverly using their implicit preference alignment and TACA mechanisms.
3. The proof looks good and is backed well by experimental results.

neg:

1. How are the implicit preference pairs actually built? If it just relies on simple heuristics (like adding noise) to define what's "better" or "worse," the model might only learn to fix low-level textures instead of understanding real structural defects. We need to see visual examples of these pairs.
2. Is TACA too rigid?  If TACA uses a fixed schedule rather than adapting dynamically to the defect type, it might just be overfitting to MVTec's specific distributions.

---

> ### Author Rebuttal · Authors · 2026-03-28
>
> Thanks for your insightful and constructive comments.
>
> **Reponse to Weaknesses:**
>
> 1. The implicit preference pairs are built per real anomaly: for the same anomaly and the same noisy latent, we form a pair from the denoising prediction of the current policy model and that of the frozen base model. The one that recovers the real anomaly pattern more faithfully is treated as preferred.
>
>     APO does not build preference pairs from heuristic perturbations. Instead, for the same real anomaly, we compare the denoising predictions of the current policy model and the frozen base model on the same noisy latent. If the policy model recovers the anomaly pattern more faithfully, that comparison is treated as preferred. So “better” is defined by which model explains the same real defect more accurately, not by the added noise itself. The noise is only the standard diffusion variable used to probe the denoising process, which is why APO learns structural defects rather than merely refining low-level textures.
>
>    Fig. 2 provides visual evidence: for capsule squeeze and hazelnut crack, APO preserves the overall object shape while introducing realistic local defect deformations, whereas the baselines often blur details or distort the whole object.
>
> 2. TACA is not a completely fixed or dataset-specific rule. Instead, it is a dynamic function strictly conditioned on the diffusion timestep $t$, as formulated in Eq. 16. By leveraging the temporal dynamics of diffusion models (high noise for structure, low noise for texture), TACA avoids overfitting to MVTec's specific distributions.
>
>    We further tested two variants: removing TACA, and using an MLP that adapts the allocation dynamically to the defect condition. The dynamic version did not improve performance and instead degraded results, while removing TACA also consistently hurt performance. This suggests that, in the few-shot setting, defect-type-adaptive scheduling is more prone to overfitting limited anomaly data, whereas our timestep-based design is more robust.
>    | Method Variant |MVTec IC-LPIPS | MVTec AP-P (%) | VisA IC-LPIPS | VisA AP-P (%) |
>    |-------|-------|--------|---------|------|
>    |APO w/o TACA| 0.35| 83.4| 0.22| 61.5|
>    |APO w/ Dynamic TACA| 0.37| 85.1| 0.25|64.2|
>    |APO w/ Fixed TACA (Ours) |**0.41**|**87.7**|**0.28**|**67.3**|
>
>
> **Reponse to Key Questions:**
>
> 1.  The primary heuristic used  relates to the temporal dynamics of diffusion models: high-noise timesteps typically govern global structures but carry a high risk of background overfitting, while low-noise steps are crucial for refining fine textures.
> While this coarse-to-fine intuition is a recognized prior, our innovation lies in uniquely integrating this temporal heuristic as a dynamic capacity constraint directly into the preference learning process. Rather than using it as a post-hoc sampling trick, we mathematically formalize it through TACA as a strict implementation of our KL divergence constraint (Eq. 8). This allows us to explicitly control the distribution shift during optimization, ensuring precise anomaly adaptation while preserving normal background structures. We clarify this in the revised manuscript
>
>
> 2.  TACA does not dynamically adjust to the specific anomaly type. Instead, its schedule is predefined as a function of the diffusion timestep (Eq. 16), not conditioned on anomaly identity. This is because TACA is designed around the role of diffusion stages themselves: high-noise steps mainly control global structure, while low-noise steps mainly refine local defect details. In this sense, TACA is timestep-aware rather than anomaly-specific, and we clarify this more explicitly in the revision.

---

> > ### Author Rebuttal · Reviewer_zqBA · 2026-04-04
> >
> > The experimental results in the rebuttal solve my concerns.

---

> > > ### Author Response · Authors · 2026-04-04
> > >
> > > Dear Reviewer zqBA,
> > >
> > > We sincerely appreciate your ongoing engagement with our paper and your constructive follow-up comments. Thank you for taking the time to carefully review our rebuttal and the newly incorporated experimental data.
> > >
> > > Your insightful feedback has been invaluable in helping us refine the clarity and overall quality of our work. We are committed to updating the manuscript to reflect your suggestions, as well as the feedback provided by the other reviewers.
> > >
> > > Thank you again for your dedicated time and thoughtful assessment.
> > >
> > > Sincerely,
> > >
> > > Authors

---

### Decision · Program_Chairs · 2026-04-30

**Decision:**

Accept (regular)

**Comment:**

This paper treat few-shot image generation pipeline casting as preference learning problem. The proposed method achieved practical balancing  between realism and diversity. In addition, time-aware allocation module and hierarchical sampling were also proposed to accelerate inferences.

Reviewers had a common positive feedbacks : well supported claim by huge experiments and technical soundness.

On the other hand, several weakness points were summarized as follows : unclear pairing method (zqBA), clarity about motivation and formulation(5h3t, Jcv8, pCGf), lacks of benchmark set (Jcv8).
After followed rebuttal responses, the most of weakness were resolved leading to all positive evaluation from 1WA, 3WR => 4WA. In detail, the authors explained pairing method clearly, clarity about motivation and formulation answer partially, and diverse benchmark sets were properly compared under additional experiments.

Given this clear consensus among reviewers, I recommend accepting this paper. When camera ready phase, please include response's table results into main or supplementary materials.